# Differentiable Structure Learning with Partial Orders

**Taiyu Ban**   **Lyuzhou Chen**   **Xiangyu Wang**[*]   **Xin Wang**   **Derui Lyu**   **Huanhuan Chen**[*]

University of Science and Technology of China
{banty, clz31415, wz520, drlv}@mail.ustc.edu.cn
{sa312, hchen}@ustc.edu.cn

## Abstract

Differentiable structure learning is a novel line of causal discovery research that transforms the combinatorial optimization of structural models into a continuous optimization problem. However, the field has lacked feasible methods to integrate partial order constraints, a critical prior information typically used in real-world scenarios, into the differentiable structure learning framework. The main difficulty lies in adapting these constraints, typically suited for the space of total orderings, to the continuous optimization context of structure learning in the graph space. To bridge this gap, this paper formalizes a set of equivalent constraints that map partial orders onto graph spaces and introduces a plug-and-play module for their efficient application. This module preserves the equivalent effect of partial order constraints in the graph space, backed by theoretical validations of correctness and completeness. It significantly enhances the quality of recovered structures while maintaining good efficiency, which learns better structures using 90% fewer samples than the data-based method on a real-world dataset. This result, together with a comprehensive evaluation on synthetic cases, demonstrates our method's ability to effectively improve differentiable structure learning with partial orders.

## 1   Introduction

Learning directed acyclic graph (DAG) structures from observational data is fundamental for causal discovery in scientific research [Opgen-Rhein and Strimmer, 2007; Pearl and others, 2000]. Traditionally, it has been approached as a combinatorial optimization problem dominated by independence tests and score-and-search methods [Heinze-Deml *et al.*, 2018]. Zheng *et al.* [2018] reformed it as a continuous optimization problem through a novel characterization of the acyclicity constraint in a differentiable form. Subsequently, numerous studies have proposed various learner architectures [Yu *et al.*, 2019; Zhu *et al.*, 2019; Zheng *et al.*, 2020], acyclicity characterizations [Yu *et al.*, 2019; Ng *et al.*, 2022; Bello *et al.*, 2022], and optimization techniques [Wei *et al.*, 2020; Deng *et al.*, 2023a,b] to advance differentiable structure learning.

In practical scenarios of causal discovery, researchers often possess prior knowledge of ordering, such as known gene activation sequences in genetics [Olson, 2006], standard treatment sequences in healthcare management [Denton *et al.*, 2007], and sequential seasonal weather patterns in meteorology [Bruffaerts *et al.*, 2018]. Such prior information can be generally formalized as a set $\mathcal{O} = \{(x, y) \mid x, y \in X\}$ of partial orders, where the binary relation $(x, y)$ represents that variable $x$ precedes $y$ in the ordering, denoted as $x \prec y$. For traditional score-and-search structure learning methods, the constraints of partial orders can reduce the space of total orderings in which the search algorithms[2] are usually performed [Teyssier and Koller, 2005]. Thus, the prior partial order informs the search process to find more genuine structures, which is essential for practical causal discovery.

---

[*]Corresponding authors.

[2]Such algorithms are known as order-based search in the literature, see Appendix A for details.

38th Conference on Neural Information Processing Systems (NeurIPS 2024).

However, the application of prior partial orders in the context of differentiable structure learning has not been explored. The main challenge arises from the inapplicability of the scenario, as the continuous optimization of structures is conducted in the graph space, while partial orders are constraints in the ordering space, leading to a misalignment of the hypothesis space. In related studies, Deng *et al.* [2023a] use a search strategy to solve the constrained optimization problem in the ordering space, which is not a purely continuous method. Some research applies differentiable structure learning to dynamic Bayesian network (DBN) structure learning [Pamfil *et al.*, 2020; Sun *et al.*, 2023; Yang *et al.*, 2022], which assumes a strict time-series ordering. This strict order prior can be simply implemented in the graph space by freezing the parameters of certain edges (see discussions in Appendix B), which significantly differs from the general partial orders discussed in this paper.

Despite of the misalignment of hypothesis spaces, the partial order constraint in the ordering space has an equivalent form in the graph space captured by path prohibitions [Grimmett and Stirzaker, 2015]. This leads to a feasible way to apply partial orders in differentiable structure learning. However, for a sequential ordering with $m$ variables, there are $\binom{m}{2}$ paths to be forbidden in the equivalent constraint. This complexity makes it impractical to develop constraints on path prohibitions individually, leading to substantial computational overhead for long sequential orderings.

To address this issue, we propose an efficient approach that augments the acyclicity constraint to naturally forbid all paths in the equivalent constraint of partial orders $\mathcal{O}$. Concretely, we infer the transitive reduction of $\mathcal{O}$ and divide it into maximal paths to capture all possible sequential orderings. These paths are then individually added to the adjacency matrix of the acyclicity term, forming an augmented acyclicity constraint. We prove that adherence to this new constraint is equivalent to adherence to partial order constraints. Furthermore, this method efficiently handles long sequential orderings, requiring only one factor to describe a sequential ordering regardless of its length. It is a plug-and-play module that can be easily adapted to various algorithms. Evaluations on both synthetic and real-world data verify its effectiveness. Contributions are listed:

- To the best of our knowledge, this is the first work to discuss the integration of prior constraints of partial orders into the continuous optimization of structure learning. We propose a plug-and-play module enabling the integration of this prior, which is theoretically applicable to all continuous methods in the differentiable structure learning context.

- We address the misalignment between the hypothesis space of differentiable structure learning and partial order constraints by converting them into an equivalent form of path prohibition constraints. By formalizing the continuous characterization of this equivalent constraint, we show its limited practicality in dealing with long sequential orderings.

- To efficiently integrate long sequential orderings, we introduce a novel approach to apply path prohibitions by augmenting the acyclicity constraint with partial orders. We prove the equivalence of this augmented acyclicity constraint to the adherence to partial orders and show its efficiency in dealing with long sequential orderings.

## 2 Notations and Preliminaries

**Notations** In the following illustrations, we denote $W_{i,:}$, $W_{:,j}$, and $W_{i,j}$ to represent the $i$th row, $j$th column, and $(i, j)$th element of a matrix $W$, respectively. If a matrix symbol includes a subscript, like $W_s$, we represent its elements as $W_{s,i,j}$. For operations resulting in a matrix, such as $W_1 + W_2$, we denote elements of the resulting matrix as $(W_1 + W_2)_{i,j}$. For simplicity, the symbol $(i, j)$ is used contextually: it can refer to a partial order $X_i \prec X_j$ or to a directed edge $(X_i, X_j)$ in the graph. Related work and proof of statements can be found in Appendix A and Appendix C, respectively.

**Structural equation model** Let $G$ denote a directed acyclic graph (DAG) with $d$ nodes, where the vertex set $V$ corresponds to a set of random variables $X = \{X_1, X_2, \ldots, X_d\}$, and the edge set $E(G) \subset V \times V$ defines the causal relationships among the variables. The structural equation model (SEM) specifies that the value of each variable is determined by a function of its parent variables in $G$ and an independent noise component:

$$X_j = f_j(\mathrm{Pa}_j^G, z_j) \tag{1}$$

where $\mathrm{Pa}_j^G = \{X_i \mid X_i \in X, (X_i, X_j) \in E\}$ denotes the set of parent variables of $X_j$ in $G$, and $z_j$ represents noise that is independent across different $j$. Denoting the structure of $G$ as a weighted

adjacent matrix $W \in \mathbb{R}^{d \times d}$, where $W_{i,j} \neq 0$ equals that $(X_i, X_j) \in E(G)$, we have:

$$X_j = f_j(W_{:,j}, X, z_j) \tag{2}$$

Given a set of samples $D \in \mathbb{R}^{m \times d}$ generated from this model with either linear or nonlinear functions in $\{f_j\}$, we next describe the process of learning the structure of $G$ in a differentiable manner.

**Differentiable structure learning** The objective of structure learning is to deduce the DAG structure represented by the weighted adjacency matrix $W \in \mathbb{R}^{d \times d}$ from the data $D$ generated by a specific set of functions $f$. We define all parameters characterizing $W$ and $f$ as $\theta$ and the graph as $G(W(\theta))$, and we formalize the optimization problem of structure learning as follows:

$$\min_{\theta} \mathcal{F}(D, f_\theta(D)) \quad \text{subject to } G(W(\theta)) \in \text{DAG} \tag{3}$$

where $\mathcal{F}$ is the score function, such as the least squares $\mathcal{F}(D, f_\theta(D)) = \frac{1}{2m} \sum_{i=1}^{m} \|D - f_\theta(D)\|_F^2$ [Loh and Bühlmann, 2014]. For clarity, we emphasize the target $W$ and rewrite it as $F(W)$. Similarly, the symbol $G(W(\theta))$ denoting the graph is simplified as $G(W)$. The acyclicity degree of the graph can be characterized by a series of continuous functions for a non-negative matrix $B$:

$$h(B) = \text{Trace}\left(\sum_{i=1}^{d} c_i B^i\right), \quad c_i > 0 \tag{4}$$

For this acyclicity characterization, Zheng *et al.* [2018] use $h(B) = \text{Trace}(e^B) - d$, derived from an infinite power series[3] of $B$. Yu *et al.* [2019] suggest a polynomial form $h(B) = \text{Trace}((I + \frac{1}{d}B)^d - I)$, and Bello *et al.* [2022] employ a log-determinant function $h(B) = -\log \det(sI - B) + d \log s$. For $W \in \mathbb{R}^{d \times d}$, it is common to use $B = W \circ W$ to ensure the non-negativity of $B$. Hence, the following mentioned $h(W)$ actually refers to $h(W \circ W)$.

**Proposition 1.** *(Theorem 1 in [Wei* et al.*, 2020]). The directed graph of a non-negative adjacency matrix $B$ is a DAG if and only if $h(B) = 0$ for any $h$ defined by* (4).

According to this result, the constraint $G(W) \in \text{DAG}$ can be implemented by the continuous equality $h(W) = 0$. This transformation converts the structure learning problem into a continuous optimization problem with an equality constraint, formulated as:

$$\min_{W \in \mathbb{R}^{d \times d}} \mathcal{F}(W) \quad \text{subject to } h(W) = 0 \tag{5}$$

Zheng *et al.* [2018] apply the augmented Lagrangian method to solve this problem, a technique widely adopted in subsequent studies. Note that our proposed plug-and-play module does not alter the optimization process; therefore, this aspect is out of scope and not discussed in this paper.

**Role of orders in structure learning** Variable ordering plays a crucial role in the combinatorial optimization of structure learning. The order-based score-and-search method is a critical research direction in this context. It is founded on the principle that structure learning is no longer NP-hard when the total ordering of variables is known [Teyssier and Koller, 2005]. These methods search within the hypothesis space of total orderings to identify the optimal solution within this constraint [Xiang and Kim, 2013; Raskutti and Uhler, 2018; Squires *et al.*, 2020; Wang *et al.*, 2021; Solus *et al.*, 2021; Chen *et al.*, 2019; Li and Beek, 2018; Chen *et al.*, 2016]. They benefit from prior partial orders, which help reduce the space of possible total orderings. However, differentiable structure learning solves Equation (5) in the DAG space, making it inapplicable to directly use partial orders. Therefore, we employ an alternative constraint equivalent to partial orders in the following section.

## 3 Differentiable Structure Learning with Partial Orders

This section introduces the integration of partial order constraints into the continuous DAG optimization framework. First, we convert the constraints from the ordering space into an equivalent form in the DAG space. Next, we discuss the limitations of direct characterizations of these equivalent constraints. Finally, we present an efficient approach to integrate partial order constraints and illustrate its theoretical correctness and completeness.

---

[3]The terms with powers higher than $d$ are expressible in finite terms with powers not exceeding $d$ by the Cayley-Hamilton theorem. Therefore, $\text{Trace}(e^B) - d$ is also an instance of Equation (4).

## 3.1 Capture partial orders with path prohibition constraints

To begin with, we formally define critical concepts related to partial orders.

**Definition 1** (Partial Order). *For a set $S$ of variables, a partial order is a binary relation $\prec$ on $S$ which is a subset of $S \times S$. For all elements $x, y$, and $z$ in $S$, the following properties are satisfied:*

*Reflexivity: $x \prec x$ for every $x$ in $S$; Antisymmetry: If $x \prec y$ and $y \prec x$, then $x = y$; Transitivity: If $x \prec y$ and $y \prec z$, then $x \prec z$.*

For the structure, if $x \prec y$, then $y$ cannot be the ancestor of $x$; that is, no directed path exists from $y$ to $x$ in the graph. Note that while the partial order relation is transitive, the absence of paths is not. This requires further consideration of the transitive property of orders.

**Definition 2** (Transitive closure). *For a set $S$ and a binary relation $\mathcal{R} \subseteq S \times S$, the transitive closure of $\mathcal{R}$, denoted by $\mathcal{R}^+$, is defined as $\mathcal{R}^+ = \bigcup_{n=1}^{\infty} \mathcal{R}^n$. $\mathcal{R}^n$ is defined recursively by: $\mathcal{R}^1 = \mathcal{R}$, $\mathcal{R}^{n+1} = \mathcal{R} \circ \mathcal{R}^n$, $\mathcal{R} \circ \mathcal{T} = \{(x, z) \in S \times S \mid \exists y \in S \text{ such that } (x, y) \in \mathcal{S} \text{ and } (y, z) \in \mathcal{T}\}$.*

**Remark 1.** *The transitive closure $\mathcal{O}^+$ of a set of partial orders $\mathcal{O}$ encompasses all orders either directly contained in or inferable through transitivity from $\mathcal{O}$.*

Now, we consider the following result from graph theory, which is essential for transforming order constraints into structural constraints.

**Proposition 2.** *There exists at least one topological sort of DAG $G$ that satisfies the partial order set $\mathcal{O}$ if and only if, for any order $(i, j)$ in $\mathcal{O}^+$, $X_j$ is not an ancestor of $X_i$ in $G$.*

With this statement, the structure learning problem with partial orders $\mathcal{O}$ can be implemented by its equivalent constraint set of path prohibitions, formalized as:

$$\min_{W \in \mathbb{R}^{d \times d}} \mathcal{F}(W) \quad \text{subject to } h(W) = 0, \ X_j \rightsquigarrow X_i \notin G(W) \text{ for all } (i, j) \in \mathcal{O}^+ \tag{6}$$

where $X_j \rightsquigarrow X_i \notin G(W)$ indicates that no directed path exists from $X_j$ to $X_i$ in $G(W)$. Subsequently, we introduce this constraint's continuous characterization and discuss its limitations.

## 3.2 Continuous characterization of path prohibitions

This section introduces the continuous characterization of the path prohibition constraint in Equation (6) and the practical difficulties in optimizing it. To clarify the unique challenges when dealing with flexible partial orders, we begin with the case of total orderings.

**Definition 3** (Total Ordering). *A total ordering is a permutation $\pi$ of all the variables, with $\pi(i)$ denoting the index of the variable in the $i$th position. $X_{\pi(i)}$ precedes $X_{\pi(j)}$ if and only if $i < j$.*

For total ordering, the order relationship between any pair of variables is contained in the transitive closure $\pi^+$. This property allows for a simple implementation of the constraint of $\pi$ by edge prohibition, as illustrated below:

**Proposition 3.** *A graph $G$ is a DAG and satisfies total ordering $\pi$ if and only if edge $(u, v)$ does not exist in $G$ for all $(v, u) \in \pi^+$.*

This edge absence constraint on $G(W)$ can be directly implemented by setting the corresponding parameters in $W$ to zero, resulting in the following formulation:

$$\min_W \mathcal{F}(W) \quad \text{subject to } W_{\pi(i), \pi(j)} = 0 \text{ for all } i \geq j \tag{7}$$

In this case, structure learning becomes an unconstrained optimization problem, as adherence to total orderings naturally satisfies the DAG constraint. This problem can be solved more efficiently than the original problem with the constraint equality $h(W) = 0$.

Now we consider the flexible partial order constraints. Let $\mathcal{O}$ represent a set of partial orders that do not inherently contain cycles within their transitive closure $\mathcal{O}^+$. Merely forbidding edges that violate $\mathcal{O}^+$ is insufficient for compliance, as it is possible to *walk* from a variable to a preceding variable in $\mathcal{O}$ through another variable whose order with others is not contained in $\mathcal{O}$, such as:

**Example 1.** *For example, consider four nodes $1, 2, 3, 4$ with a partial order set $\mathcal{O} = \{(1, 2), (2, 3)\}$. We forbid all inverse edges in $\mathcal{O}^+$, which are $(2, 1), (3, 2), (3, 1)$. Despite this, directed paths violating the partial order $(1, 2)$ can still exist, such as the path $(2, 4, 1)$. Such paths can be constructed by traversing nodes not in $\mathcal{O}$, like node 4 in this case.*

Consequently, we must consider the constraint of path prohibitions equivalent to $\mathcal{O}$. According to the proof to Proposition 1, the following equality can be used for path prohibition constraints.

**Proposition 4.** *No directed path $X_i \rightsquigarrow X_j$ exists in $G(W)$ if and only if $\left( \sum_{l=1}^{d} (W \circ W)^l \right)_{i,j} = 0$.*

With this statement, we can formalize the optimization problem in Equation (6) as follows:

$$\min_W \mathcal{F}(W) \quad \text{subject to } h(W) = 0, \ p(W, \mathcal{O}) = 0 \tag{8a}$$

$$p(W, \mathcal{O}) = \sum_{(i,j) \in \mathcal{O}^+} \left( \sum_{l=1}^{d} (W \circ W)^l \right)_{j,i} \tag{8b}$$

**Remark 2.** *A significant difficulty of the optimization problem formulated in Equation (8a) is its steep decline in training efficiency as the complexity of partial orders increases. The penalty term $p(W, \mathcal{O})$, as defined by Equation (8b), includes a term for each order in $\mathcal{O}^+$, directly impacting the computational cost for gradient calculations. When dealing with a sequential ordering with $m$ variables, it introduces $\binom{m}{2}$ new terms. Each of these terms demands comparable time for gradient calculation to the acyclicity term $h(W)$ typically used in current studies. This makes the computational load impractical for long sequential orderings. Note that the total ordering constraint results in the most constraint terms in this case, while it can be efficiently addressed by Equation (7).*

*This observation underpins the need to develop a more efficient method to ensure that the structure learning process remains computationally feasible for long sequential orderings.*

### 3.3 Augmented acyclicity-based partial order characterization

This section introduces an efficient method to characterize partial orders, distinct from directly representing the equivalent path prohibitions. We first introduce some critical concepts.

**Definition 4** (Transitive Reduction). *The transitive reduction $\mathcal{O}^-$ of a relation $\mathcal{O}$ is the smallest relation such that the transitive closure of $\mathcal{O}^-$ is equal to the transitive closure of $\mathcal{O}$. Formally, $(\mathcal{O}^-)^+ = \mathcal{O}^+$ and $\mathcal{O}^-$ is minimal.*

The transitive reduction is used to eliminate redundant orders to facilitate calculation efficiency. Below, we provide an example to illustrate transitive reduction alongside transitive closure.

**Example 2.** *For a set of transitive binary relation $\mathcal{O} = \{(1, 2), (2, 3), (1, 3), (3, 4)\}$, its transitive closure is $\mathcal{O}^+ = \mathcal{O} \cup \{(1, 4), (2, 4)\}$, and its transitive reduction is $\mathcal{O}^- = \mathcal{O} \setminus \{(1, 3)\}$.*

**Definition 5.** *Let $G = (V, E)$ be a graph. A source is a vertex in $V$ with no incoming edges, i.e., $\{v \in V : \deg^-(v) = 0\}$. A sink is a vertex with no outgoing edges, i.e., $\{v \in V : \deg^+(v) = 0\}$.*

**Definition 6** (Maximal Path). *Let $G = (V, E)$ be a graph with a node set $V$ and edge set $E$. A path $p = (v_1, \ldots, v_k)$ with $(v_i, v_{i+1}) \in E$ is considered a maximal path if $v_1$ is a source, $v_k$ is a sink, and the path is not a proper subsequence of any other path from $v_1$ to $v_k$.*

**Definition 7.** *The transitive closure of a path $p = (v_1, \ldots, v_k)$, denoted as $p^+$, is the set of all ordered pairs $(v_i, v_j)$ for $1 \leq i < j \leq k$.*

**Remark 3.** *For brevity, the following discussions regard the concepts of the directed graph, path, sequential ordering, and partial order set as an identical type of set whose element is an ordered pair $(i, j)$, as both node reachability in a graph and the order relationship are transitive. Especially, we do not distinguish between a partial order set $\mathcal{O}$ and the graph $G(\mathcal{O})$ constructed by $E(G(\mathcal{O})) = \{(i, j) \mid (i, j) \in \mathcal{O}\}$.*

**Remark 4.** *We assume that no cycle exists in $\mathcal{O}^+$. That is, $\mathcal{O}$ is not conflicting with itself.*

With these definitions, we formalize the new approach to integrating partial order constraint $\mathcal{O}$ into differentiable structure learning as follows (see Appendix D for detailed implementations):

$$\min_W \mathcal{F}(W) \quad \text{subject to} \quad h'(W, \mathcal{O}) = 0 \tag{9a}$$

$$h'(W, \mathcal{O}) = \sum_{o \in \mathcal{P}(\mathcal{O}^-)} h(\mathcal{A}(W, o)) \tag{9b}$$

$$\mathcal{A}(W, o) = W + \tau W_o - W \circ W_o \tag{9c}$$

$$W_{o,i,j} = [(i, j) \in o] \tag{9d}$$

Here, $\mathcal{O}^-$ is the transitive reduction of $\mathcal{O}$. $\mathcal{P}(\mathcal{O}^-)$ represents the set of all maximal paths of $\mathcal{O}^-$. $[P]$ is the indicator function valuing 1 if condition $P$ holds and 0 otherwise. $\tau > 0$ is a hyperparameter used for adjusting the weight in gradient calculation.

**Remark 5.** *Recall that $h(W) \geq 0$ by Equation (4). Then we have that $h'(W, \mathcal{O}) = 0$ is equivalent to $h(\mathcal{A}(W, o)) = 0$ for $o \in \mathcal{P}(\mathcal{O}^-)$ by Equation (9b).*

Equation (9) can be interpreted as augmenting the original acyclicity constraint $h(W) = 0$ to a *stronger* one $h'(W, \mathcal{O}) = 0$. Specifically, we use a series of partial order-augmented acyclicity constraints $h(\mathcal{A}(W, o)) = 0$ for $o$ in the maximal path set of $\mathcal{O}^-$ as described in Equation (9b). For each augmented acyclicity, we add the path $o$ to the adjacency matrix $W$ by $\mathcal{A}(W, o)$ as detailed in Equation (9c). Thus, the acyclicity function $h$ with $\mathcal{A}(W, o)$ as input represents a *stronger* acyclicity constraint. The *additional* part of this stronger acyclicity accurately captures adherence to the sequential ordering indicated by $o$, which can be derived from the following statement.

**Lemma 1.** *A graph $G$ is a DAG and satisfies a sequential ordering $o = \{(p_1, p_2, \cdots, p_m)\}$ if and only if graph $G'$ is a DAG where $E(G') = E(G) \cup o$.*

This lemma states the equivalence of $h(\mathcal{A}(W, o)) = 0$ to adherence to the sequential ordering $o$. Now consider the following statement.

**Lemma 2.** *For the set $\mathcal{P}(\mathcal{O}^-)$ of all maximal paths of $\mathcal{O}^-$, the union of the transitive closures of these paths is the transitive closure of $\mathcal{O}$: $\bigcup_{o \in \mathcal{P}(\mathcal{O}^-)} o^+ = \mathcal{O}^+$*

This lemma states that adherence to all the sequential orderings $o$ indicated by maximal paths in $\mathcal{O}^-$ is equivalent to adherence to the complete set $\mathcal{O}$ of partial orders. Recall that $h'(W, \mathcal{O}) = 0$ is equivalent to $h(\mathcal{A}(W, o)) = 0$ for $o$ in $\mathcal{P}(\mathcal{O}^-)$, and $h(\mathcal{A}(W, o)) = 0$ is equivalent to adherence to $o$. Hence, we derive that $h'(W, \mathcal{O}) = 0$ is equivalent to adherence to $\mathcal{O}$ by Lemma 2, as described in the following statement (the proof of these statements is provided in Appendix C.1).

**Theorem 1.** *A graph $G$ is a DAG and satisfies a set of partial orders $\mathcal{O}$ if and only if $h'(W, \mathcal{O}) = 0$ for the function $h$ defined by Equation (4) and $h'$ defined by Equations (9b), (9c), and (9d).*

Theorem 1 shows the correctness and completeness of the equality $h'(W, \mathcal{O}) = 0$ in capturing the partial order constraint $\mathcal{O}$. More concretely, all prior information of $\mathcal{O}$ is fully integrated while no extra information beyond $\mathcal{O}$ is introduced. This is attributed to two critical steps in integrating $\mathcal{O}$ into the acyclicity constraint. **Step 1.** Split the partial order constraint $\mathcal{O}$ into sequential orderings. **Step 2.** Ensure that these sequential orderings are maximal paths in $\mathcal{O}^-$. If **Step 1** is removed[4], and we directly add all the edges in $\mathcal{O}^-$ to $G(W)$ for augmented acyclicity, $h'$ will degenerate into $h(\mathcal{A}(W, \mathcal{O}^-))$. This introduces extra orders outside of $\mathcal{O}$, as indicated in the following example.

**Example 3.** *Assume that $h'(W, \mathcal{O}) \equiv h(\mathcal{A}(W, \mathcal{O}^-))$. Consider a partial order set $\mathcal{O} = \{(1, 2), (3, 4)\}$ and a DAG $G(W)$ with edges $E(G(W)) = \{(2, 3), (4, 1)\}$. Obviously, $G(W)$ satisfies $\mathcal{O}$. Consider the graph constructed by adding edges in $\mathcal{O}^-$ (where $\mathcal{O}^- = \mathcal{O}$) to $G(W)$, i.e., the graph of the matrix $\mathcal{A}(W, \mathcal{O}^-)$. Its edge set is $\mathcal{O} \cup E(G(W)) = \{(1, 2), (2, 3), (3, 4), (4, 1)\}$ and contains a cycle $(1, 2, 3, 4, 1)$. This makes $h'(W, \mathcal{O}) \equiv h(\mathcal{A}(W, \mathcal{O})) \neq 0$ by Proposition 1.*

In this case, a *legal* DAG that satisfies $\mathcal{O}$ is forbidden by the constraint $h'(W, \mathcal{O}) = 0$, indicating that extra constraints beyond the prior are introduced. In other words, **Step 1** guarantees the *necessity* of the constraint equality for partial order $\mathcal{O}$. For **Step 2**, it guarantees the *sufficient* adherence to $\mathcal{O}$. If this step is removed, some order constraints of $\mathcal{O}$ can be lost. See the following case.

---

[4]Note that Step 2 would also be omitted in the absence of Step 1.

**Example 4.** *Consider a partial order constraint set $\mathcal{O} = \{(1,2),(2,3),(4,2)\}$. Suppose that it is divided into two sequential orderings $o_1 = (1,2,3)$ and $o_2 = (4,2)$, where $o_2$ is not the maximal path of $\mathcal{O}^-$. Then consider graph $G(W)$ with edges $\{(1,2),(3,4)\}$. We have that $E(G(\mathcal{A}(W,o_1))) = \{(1,2),(2,3),(3,4)\}$ and $E(G(\mathcal{A}(W,o_2))) = \{(1,2),(3,4),(4,2)\}$. Both graphs are DAGs satisfying $h(\mathcal{A}) = 0$ by Proposition 1. Then we have $h'(W,\mathcal{O}) \equiv h(\mathcal{A}(W,o_1)) + h(\mathcal{A}(W,o_2)) = 0$. Even if $G(W)$ satisfies this constraint equality, the edge $(3,4)$ in $G(W)$ still violates the order $(4,3) \in \mathcal{O}^+$, derived by the transitive result of $(4,2)$ and $(2,3)$.*

In this case, an *illegal* instance that violates $\mathcal{O}$ is not forbidden by the constraint $h'(W,\mathcal{O}) = 0$. This indicates that some orders in $\mathcal{O}^+$ are not specified by $h'(W,\mathcal{O}) = 0$ if **Step 2** is omitted.

**Remark 6.** *Now we discuss the complexity of gradient calculation for $h'(W,\mathcal{O})$. Equation (9b) indicates that this complexity is determined by the number $|\mathcal{P}(\mathcal{O}^-)|$ of maximal paths in $\mathcal{O}^-$, rather than the size $|\mathcal{O}^+|$ of its transitive closure. For a sequential ordering with $m$ variables, $h'$ contains only one factor of $h$ regardless of the value of $m$. This addresses the impractical computational load of path prohibition constraints with $\binom{m}{2}$ factors as discussed in Remark 2. Note that the computational complexity of $h'(W,\mathcal{O})$ can increase with multiple sequential orderings, which is evaluated in the following section.*

# 4 Experiments

We evaluate our module for applying prior partial order (PPO) on linear NOTEARS [Zheng *et al.*, 2018], NOTEARS-MLP [Zheng *et al.*, 2020], and DAGMA [Bello *et al.*, 2022]. It is named as 'PPO-*alg-l-p*', where *alg* is the backbone algorithm, and *l,p* are settings on partial orders. Representative results are reported here, and the complete results are available in Appendix E.

Section 4.1 presents the results on synthetic datasets. Section 4.2 discusses the results obtained using a well-established biological dataset [Sachs *et al.*, 2005]. For computational resources, linear NOTEARS and DAGMA are executed on a 32-core AMD Ryzen 9 7950X CPU at 4.5GHz, while NOTEARS-MLP uses an NVIDIA GeForce RTX 3090 GPU, both with a 32GB memory limit. We conduct five simulations for each synthetic structure and one simulation for the data and partial order constraints for each structure.

## 4.1 Synthetic datasets

Random DAGs are generated using Erdös-Rényi (ER) and scale-free (SF) models with node degrees in $\{2,4\}$ and numbers of nodes $d$ in $\{20,30,50\}$. For linear SEM, uniformly random weights are assigned to the weighted adjacency matrix $A$. Given $A$, samples are generated by $X = A^T X + z, X \in \mathbb{R}^d$ using noise models {Gaussian (gauss), Exponential (exp)}. Observational samples $D \in \mathbb{R}^{n \times d}$ are then generated with the sample size $n = 4d$. For nonlinear cases, uniformly random weights are assigned to weighted adjacency matrices $W_1, W_2, W_3$. Based on these matrices, samples are generated using $X = \tanh(XW_1) + \cos(XW_2) + \sin(XW_3) + z$ with $z \sim \mathbf{N}(0,1)$. The sample size is set to $n = 20d$. The parameter $\tau$ in Equation (9c) is set to 1.

To mimic real-world prior partial orders, we generate multiple sequential orderings, referred to as the *chain* of ordering. Specifically, we first conduct a topological sort on the DAG to derive a total ordering $\pi$. We then randomly select $l$ chains, each denoting a sub-ordering randomly generated from $\pi$. Here, $l$ is the number of chains and $m$ is the size of each chain. We first investigate the case of a single chain of ordering and then the case of multiple chains of orderings. For single-chained ordering, $l = 1$ and $m$ is in $\{0.5d, 0.75d, d\}$ (where $d$ is the number of nodes). For multi-chained ordering, $l$ is in $\{1,2,3,5,10\}$ and $m$ is fixed at $0.5d$.

### 4.1.1 Single-chained ordering

In this experiment, we examine structure learning using single-chained ordering. The results of Structural Hamming Distance (SHD), F1 score, and run time for linear NOTEARS are illustrated in Figure 1. Results for NOTEARS-MLP (nonlinear samples) and DAGMA are presented in Figure 2.

**Output quality.** Our method demonstrates notable superiority over algorithms without prior information in terms of output quality in most cases, with the advantage becoming more pronounced

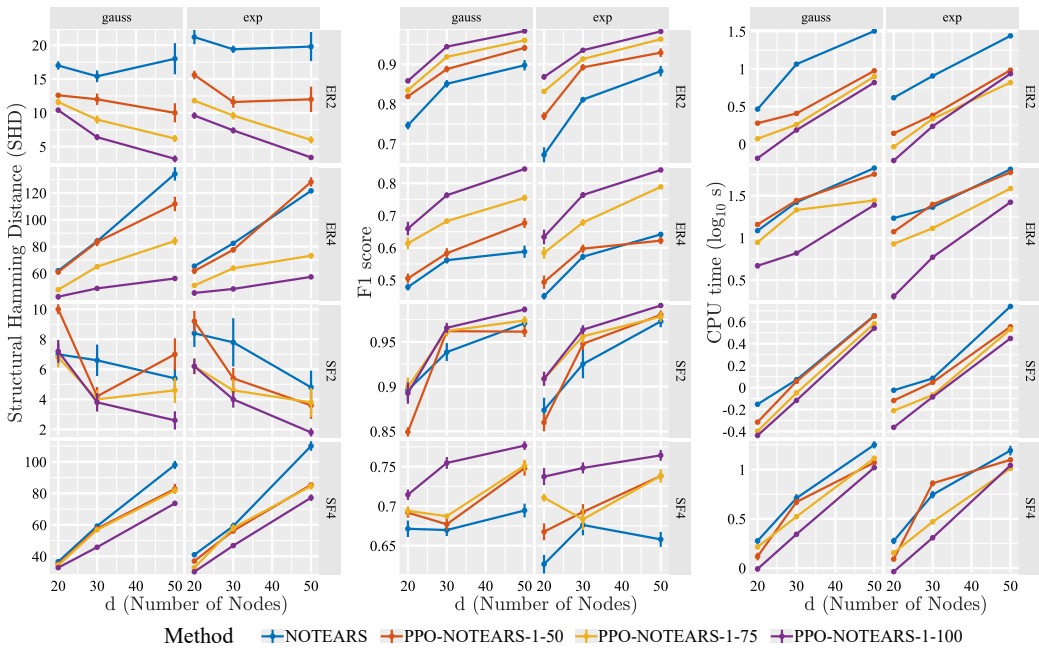

Figure 1: Structural discovery in terms of SHD (lower is better), F1-score (higher is better) and CPU time ($\log_{10}$ s) with NOTEARS on linear data. Rows: graph types. {ER,SF}-$k$ represents {Erdös-Rényi, scale-free} graphs with $kd$ expected edges. Columns: noise types of SEM. Error bars represent standard errors over 5 simulations. Method: PPO-$alg$-1-$p$ denotes our method with partial order settings $l = 1$ and $m = p\%d$.

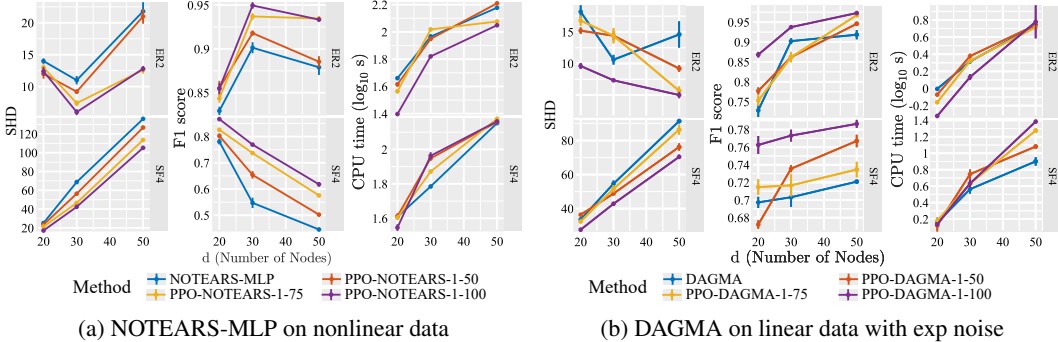

(a) NOTEARS-MLP on nonlinear data          (b) DAGMA on linear data with exp noise

Figure 2: Structural discovery in terms of SHD (lower is better), F1-score (higher is better) and CPU time ($\log_{10}$ s) with NOTEARS-MLP and DAGMA on representative cases.

as the number of nodes in the ordering chain increases. This confirms the effectiveness of our approach in enhancing structure learning quality with prior partial orders. Some degradation cases may be caused by *invalid* priors in the simulated ordering. Since the topological sort for a DAG is not unique, some random ordering chains may not contribute to revealing the most essential parts of orderings, especially with smaller chain sizes.

**Run time.** We observe that our method with a single-chained ordering is typically faster than structure learning without prior. This efficiency is due to the effective management of our module for single-chained orderings. However, in some cases, such as on the SF4 graph with NOTEARS-MLP and DAGMA, the efficiency can be degraded. This indicates that the impact of partial orders on efficiency can vary with different data distributions and backbone algorithms.

### 4.1.2 Multi-chained ordering

The results for SHD, F1-score, and run time using linear NOTEARS and NOTEARS-MLP with multi-chained orderings are presented in Figure 3. The output quality demonstrates similar trends to

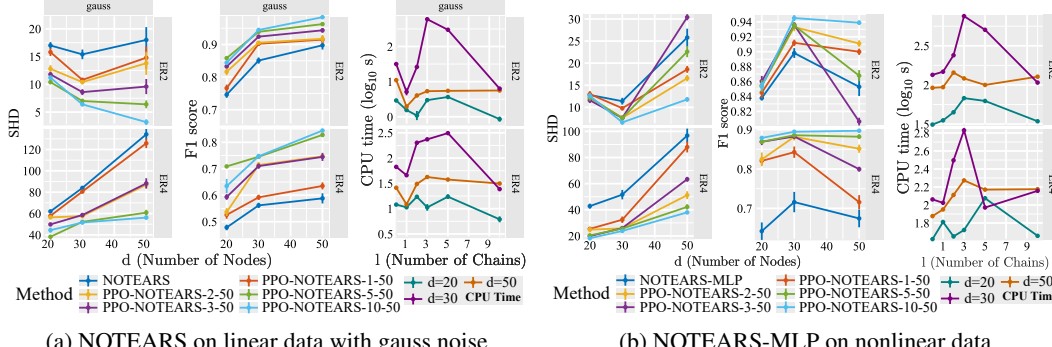

(a) NOTEARS on linear data with gauss noise      (b) NOTEARS-MLP on nonlinear data

Figure 3: Structural discovery in terms of SHD (lower is better), F1-score (higher is better) and CPU time ($\log_{10}$ s) with linear NOTEARS and NOTEARS-MLP. Method: PPO-*alg*-*l*-50 represents our method where $l$ is the number of chains and the size of chains is $m = 0.5d$.

those seen with single-chained ordering, with more pronounced improvements over the baselines as the number of partial order constraint chains increases. The run time dynamics are illustrated with a curve that reflects changes corresponding to the primary influencing factor: the number of chains. Initially, run time increases and then decreases as the number of chains grows. This pattern results from the increasing complexity of gradient calculations for the constraint term $h'$, which scales with the number of maximal paths in the partial orders. At first, the increasing number of chains leads to more maximal paths in the partial order, thus delaying time efficiency. As the partial order becomes denser, more chains can be covered by a longer chain, which leads to fewer maximal paths in its transitive reduction, resulting in better time efficiency.

Table 1: Structural discovery in terms of SHD↓, FDR↓, TPR↑ and F1↑ on sach dataset with various sample sizes. Linear NOTEARS is used as the backbone algorithm, and PPO-1-$m$ denotes our method using a single-chained ordering containing $m$ nodes. The best result is highlighted with bold texts.

| Method | sach-50 | | | | sach-100 | | | | sach-500 | | | | sach-853 | | | |
|---|---|---|---|---|---|---|---|---|---|---|---|---|---|---|---|---|
| | SHD | FDR | TPR | F1 | SHD | FDR | TPR | F1 | SHD | FDR | TPR | F1 | SHD | FDR | TPR | F1 |
| NOTEARS | 25 | 0.76 | 0.41 | 0.30 | 16 | 0.65 | 0.41 | 0.38 | 15 | 0.60 | 0.35 | 0.38 | 12 | 0.57 | 0.35 | 0.39 |
| PPO-1-6 | 20 | 0.70 | 0.35 | 0.32 | 15 | 0.62 | 0.29 | 0.33 | 13 | 0.50 | 0.35 | 0.41 | 14 | 0.55 | 0.29 | 0.36 |
| PPO-1-8 | 16 | 0.59 | 0.41 | 0.41 | 11 | 0.42 | 0.41 | 0.48 | 12 | 0.40 | 0.35 | 0.44 | 10 | 0.30 | 0.41 | 0.52 |
| PPO-1-11 | **12** | **0.27** | **0.47** | **0.57** | **10** | **0.00** | **0.41** | **0.58** | **11** | **0.13** | **0.41** | **0.56** | **11** | **0.13** | **0.41** | **0.56** |

## 4.2 Real-world data

The dataset provided by Sachs *et al.* [2005] consists of continuous measurements of protein and phospholipid expression levels in human immune system cells. It is frequently used as a benchmark in graphical models due to its associated consensus network, which includes 11 nodes and 17 edges, based on experimental annotations recognized by the biological community.

We use experimental data from one of the cells with 853 samples. To mimic varying levels of experimental resources, we selected the first $s$ data samples for testing, where $s \in \{50, 100, 500, 853\}$. A single-chained prior ordering is given involving different numbers of variables in $\{6, 8, 11\}$. Linear NOTEARS serves as the backbone algorithm. The parameter $\tau$ in Equation (9c) is set to 3.

The structural evaluation metrics reported in Table 1 include SHD, False Discovery Rate (FDR), True Positive Rate (TPR), and F1 score. The findings reveal that NOTEARS with partial orders discovers more accurate structures than NOTEARS without prior information in most cases. Remarkably, even with the smallest sample size (50), NOTEARS with partial order constraint (11 nodes) significantly outperforms the baseline using the largest sample size (853). This underscores the efficacy of the proposed partial order constraint-based differentiable structure learning approach in conserving experimental resources in scientific research contexts.

## 5 Discussion

**Limitations and future directions**  Despite the theoretical correctness and completeness of the proposed method for integrating partial orders in differentiable structure learning, there are some practical limitations.

First, the augmented acyclicity constraint $h'(W, \mathcal{O}) = 0$ cannot be strictly satisfied during the optimization process of the augmented Lagrangian method, as proven by Wei *et al.* [2020]. This may result in some order constraints from the prior not being satisfied in the output. This issue is inherent to the optimization aspect of differentiable structure learning and may be addressed with more refined optimization techniques in the future.

Additionally, although the proposed method efficiently handles long sequential orderings, its efficiency can be impacted by partial orders with complex structures. This is evident from the experimental results involving multi-chained orderings. We randomly selected multiple ordering chains, each comprising half of the nodes, forming a complex order structure with considerable maximal paths. This leads to a sharp increase in the number of constraint terms in the augmented acyclicity constraint. Fortunately, real-world ordering priors typically do not exhibit such complex structures and can usually be captured by a few chains. To enhance time efficiency in such cases, a more refined characterization method could be explored to reduce computational overhead in the future. We may focus on improving the gradient calculation of the proposed augmented acyclicity constraint in the context of multiple sequential orderings. This can be explored by merging the common parts of the gradient calculation process or developing more efficient characterizations.

**Conclusion**  This paper enhances the field of differentiable structure learning by enabling this framework to apply priors of partial order constraints. We systematically analyze the related challenges of applying flexible order constraints and propose a novel and effective strategy to address them by augmenting the acyclicity constraint. We present a theoretical proof confirming the correctness and completeness of our strategy. Empirical results highlight the superiority of our method in improving structure learning with partial order constraints. Results on a well-known real-world dataset further emphasize its potential in uncovering more accurate causal mechanisms with reduced experimental resources.

## Broader Impact

The proposed method allows researchers across various scientific fields to specify ordering priors, enhancing causal discovery with state-of-the-art differentiable structure learning algorithms from experimental or observational data. As demonstrated with the real-world Sachs dataset [Sachs *et al.*, 2005], differentiable structure learning using a proper ordering prior with only 10% of the samples required by methods without a prior yields significantly better structures. This reduction in data requirements for causal discovery can save experimental resources across many domains.

However, researchers must exercise caution when providing ordering priors. The proposed method strictly adheres to these priors, and numerous incorrect ordering priors can severely impact the results. For instance, in social sciences, if incorrect assumptions about the order of socio-economic events are used as priors, the resulting causal model may be misleading, affecting policy decisions based on such a model.

## Acknowledgements

This research was supported in part by the National Key R&D Program of China (No. 2021ZD0111700), in part by the National Nature Science Foundation of China (No. 62137002, 62176245, 62406302), in part by the Natural Science Foundation of Anhui province (No. 2408085QF195), in part by the Key Research and Development Program of Anhui Province (No. 202104a05020011), in part by the Key Science and Technology Special Project of Anhui Province (No. 202103a07020002).

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

# Appendix A   Related Work

## A.1   Combinatorial optimization of structure learning

Traditional combinatorial optimization of structure learning includes constraint-based, score-based, and hybrid methods. Constraint-based methods utilize conditional independence (CI) tests to construct the graph. The most notable example is the PC algorithm (named after its developers, Peter and Clark), which begins with a complete graph and progressively removes edges between nodes that are conditionally independent given a set of other variables [Spirtes *et al.*, 2000]. Constraint-based approaches typically result in a partially directed acyclic graph (PDAG), where some edges remain undirected due to equivalent DAG configurations.

Score-based methods employ a scoring function to assess the fit of a DAG model to the observed data, aiming to identify the graph with the best score. Commonly used scoring functions, such as BIC, BDeu, and MDL, are decomposable, which facilitates local optimization for each variable. Notable algorithms in this category include the Greedy Equivalent Search (GES) [Chickering, 2002] and the K2 algorithm [Cooper and Herskovits, 1991].

A significant subset of score-based methods assumes a total ordering of the variables to expedite finding the optimal solution within this constraint, followed by a search among different order hypotheses to locate the most fitting order. Teyssier and Koller [2005] demonstrated that with a fixed ordering, the optimal solution could be computed in polynomial time, avoiding the need for DAG consistency checks. Although searching the order space is computationally intensive, these methods have outperformed traditional DAG-based searches. Subsequent studies have expanded on this framework [Xiang and Kim, 2013; Raskutti and Uhler, 2018; Squires *et al.*, 2020; Wang *et al.*, 2021; Solus *et al.*, 2021; Chen *et al.*, 2019].

Hybrid methods integrate score-based and constraint-based approaches. They streamline the search for potential parent nodes for each variable by applying CI tests to narrow down the set of candidates. Examples of hybrid methods include Max-Min Hill Climbing (MMHC) [Tsamardinos *et al.*, 2006] and H$^2$PC [Gasse *et al.*, 2014].

## A.2   Differentiable structure learning

Zheng *et al.* [2018] introduced NOTEARS, which proposes a continuous characterization of the DAG constraint. This transforms the structure learning within the structural equation model (continuous scoring function) into a differentiable constrained optimization problem. The authors used an augmented Lagrangian method to solve this problem in a linear SEM, observing superior performance compared to state-of-the-art structure learning solvers. They further extended this approach with an MLP learner to address nonlinear SEMs [Zheng *et al.*, 2020]. Subsequent studies have advanced the field of differentiable structure learning by improving acyclicity characterization, developing more powerful neural network architectures, and enhancing optimization approaches.

Yu *et al.* [2019] formalized the structure learning task using a graphical neural network (GNN) and proposed a GNN-based structure learning approach named DAG-GNN. A major advantage of DAG-GNN is its ability to handle both discrete data (modeled by Bayesian networks) and continuous data (modeled by SEMs). Zhu *et al.* [2019] introduced a reinforcement learning approach for structure learning. Yu *et al.* [2019] used a different polynomial function to characterize acyclicity, achieving better computational efficiency. Bello *et al.* [2022] proposed DAGMA, which employs a log-determinant function, showing a stronger constraint on acyclicity. Ng *et al.* [2020] introduced a likelihood-based scoring function, demonstrating superiority over the regression-based scoring used in NOTEARS. Wei *et al.* [2020] developed a local search post-processing algorithm informed by the KKT conditions of the constrained optimization problem. Deng *et al.* [2023a] further introduced a local search differentiable structure learning algorithm based on topological swap, similar to order-based search in combinatorial structure learning.

Distinct from these studies, this paper aims to enhance the field of differentiable structure learning by integrating more types of prior constraints. Current studies typically address straightforward prior information such as constraints of edges [Wei *et al.*, 2020] and total ordering [Deng *et al.*, 2023a]. However, real-world priors are often more ambiguous and less comprehensive. Therefore,

we consider partial orders, which are general enough to represent real-world priors on ordering relationships that may be contained in experimental settings or existing domain knowledge.

## Appendix B  Discussions of Ordering Prior in DBN

This section explores the connection between dynamic Bayesian Networks (DBNs) and the ordering prior discussed in this paper, and how this prior can be implemented through parameter freezing. DBNs assume that the observational data are time-series data containing $m$ time slices. Given $d$ variables, there are $m \times d$ nodes in the DBN, where each variable has one "copy" in each time slice. The influence patterns among the variables vary according to different assumptions, with a common premise being the order constraints of the time slices [Pamfil *et al.*, 2020; Sun *et al.*, 2023; Yang *et al.*, 2022]. Specifically, a variable in a later time slice cannot affect one in an earlier time slice. This order constraint can be generally described by the following ordering type.

**Definition 8** (Time-series ordering). *In a time-series ordering $\mathcal{S}$, the entire set of variables is partitioned into distinct stages. Variables within the same stage do not have a defined ordering relative to each other, while variables in an earlier stage precede those in a later stage.*

In the time-series ordering, the partial order between any two variables in different stages is known. For a time-series ordering, it can be integrated into the structure learning by parameter freezing.

**Theorem 2.** *A graph $G(W)$ is a DAG and satisfies time-series ordering $\mathcal{S}$ if and only if $h(W) = 0$ and $W_{i,j} = 0$ for all $(j,i) \in \mathcal{S}^+$.*

*Proof.* The aim is to prove the following equivalence:

$$h(W) = 0 \text{ and } W_{i,j} = 0 \text{ for all } (j,i) \in \mathcal{S}^+ \iff G(W) \in \text{DAG and } X_i \rightsquigarrow X_j \notin G(W) \text{ for all } (j,i) \in \mathcal{S}^+.$$

**Necessity ($\Longleftarrow$):**

1. If $G(W)$ is a DAG, then by Proposition 1 in the main text, $h(W) = 0$. 2. If $X_i \rightsquigarrow X_j \notin G(W)$ for all $(j,i) \in \mathcal{S}^+$, this implies that there is no directed edge from $i$ to $j$ for any $(j,i) \in \mathcal{S}^+$. Therefore, $W_{i,j} = 0$ for all $(j,i) \in \mathcal{S}^+$.

Thus, if $G(W)$ is a DAG and $X_i \rightsquigarrow X_j \notin G(W)$ for all $(j,i) \in \mathcal{S}^+$, then $h(W) = 0$ and $W_{i,j} = 0$ for all $(j,i) \in \mathcal{S}^+$.

**Sufficiency ($\Longrightarrow$):**

1. If $h(W) = 0$, then by Proposition 1 in the main text, $G(W)$ is a DAG. 2. We need to show that $W_{i,j} = 0$ for all $(j,i) \in \mathcal{S}^+$ implies $X_i \rightsquigarrow X_j \notin G(W)$ for all $(j,i) \in \mathcal{S}^+$.

We prove this by contradiction. Assume that there exists a pair $(v,u) \in \mathcal{S}^+$ such that $X_u \rightsquigarrow X_v \in G(W)$. This means there is a directed path from $u$ to $v$ in $G(W)$.

Let the path be $(i_0, i_1, \cdots, i_{q-1}, i_q)$ where $i_0 = u$ and $i_q = v$. Since $v$ is in an earlier stage than $u$, $v$ precedes $u$ in the partial order $\mathcal{S}^+$. Therefore, there must exist at least one pair $(i_k, i_{k+1})$ in the path such that $i_{k+1}$ is in an earlier stage than $i_k$. This implies $(i_{k+1}, i_k) \in \mathcal{S}^+$, and hence $W_{i_k, i_{k+1}} \neq 0$.

However, this contradicts our assumption that $W_{i,j} = 0$ for all $(j,i) \in \mathcal{S}^+$. Therefore, our assumption is false, and it must be that $X_i \rightsquigarrow X_j \notin G(W)$ for all $(j,i) \in \mathcal{S}^+$.

Thus, if $h(W) = 0$ and $W_{i,j} = 0$ for all $(j,i) \in \mathcal{S}^+$, then $G(W)$ is a DAG and $X_i \rightsquigarrow X_j \notin G(W)$ for all $(j,i) \in \mathcal{S}^+$. $\square$

This result leads to a parameter freezing strategy for applying the time-series ordering into differentiable structure learning:

$$\min_{W \in \mathbb{R}^{d \times d}} \mathcal{F}(W) \quad \text{subject to } h(W) = 0 \text{ and } W_{i,j} = 0 \text{ for all } (j,i) \in \mathcal{S}^+ \tag{10}$$

# Appendix C  Proof of Main Results

## C.1  Proof of Theorem 1

This section proves the equivalence of the constraint equality $h'(W, \mathcal{O})$ as designed in Equation (9) to the adherence to the partial order constraint $\mathcal{O}$.

**Theorem 1.** *A graph $G$ is a DAG and satisfies a set of partial orders $\mathcal{O}$ if and only if $h'(W, \mathcal{O}) = 0$ for the function $h$ defined by Equation* (4) *and $h'$ defined by Equations* (9b), (9c), *and* (9d).

Recall the construction of $h'$: The function

$$h'(W, \mathcal{O}) = \sum_{o \in \mathcal{P}(\mathcal{O}^-)} h(\mathcal{A}(W, o))$$

where $\mathcal{P}(O^-)$ represents the set of maximal paths in $\mathcal{O}^-$. $\mathcal{A}(W, o)$ returns a adjacent matrix that represents a graph by adding the path $o$ to the graph $G(W)$, denoted as $E(G(\mathcal{A})) = E(G(W)) \cup o$. Given that $h(W) \geq 0$ by Equation (4), we have:

$$h'(W, \mathcal{O}) = 0 \iff h(\mathcal{A}(W, o)) = 0 \text{ for } o \in \mathcal{P}(\mathcal{O}^-) \tag{11}$$

The proof of Theorem 1 is completed by two statements. We consider the first one:

**Lemma 1.** *A graph $G$ is a DAG and satisfies a sequential ordering $o = \{(p_1, p_2, \cdots, p_m)\}$ if and only if graph $G'$ is a DAG where $E(G') = E(G) \cup o$.*

*Proof.* Recall the equivalent constraints of the order constraint in Proposition 4, and we have the following objective to prove:

$$G' \in \text{DAG for } E(G') = E(G) \cup o \iff G \in \text{DAG and } X_j \rightsquigarrow X_i \notin G \text{ for } (i, j) \in o^+ \tag{12}$$

($\impliedby$) We prove the necessity by contradiction. Suppose that the right side holds and $G'$ is not a DAG. Consider a cycle $(c_1, c_2, \cdots, c_k, c_1)$ in $G'$ under the following two cases.

1) No edge in the cycle is contained in $o$. Then all these edges in the cycle belong to $G$ by the condition $E(G') = E(G) \cup o$. This conflicts with the fact that $G$ is a DAG.

2) If some edges are contained in $o$ and they form a consecutive path $(c_{r_1}, c_{r_1+1}, \cdots, c_{q_1})$. In this case, the rest part of the cycle $(c_{q_1}, c_{q_1+1}, \cdots, c_{r_1})$ is contained in the graph $G$. This forms a directed path from $c_{q_1}$ to $c_{r_1}$ for $(c_{r_1}, c_{q_1}) \in o^+$, which conflicts with the condition of the right-hand side of Equality (12) $X_j \not\rightsquigarrow X_i \notin G$ for $(i, j) \in o^+$.

3) Some edges are contained in $o$ and do not form a single consecutive path. Then we can represent them as a set of disjoint paths in sequence:

$$r_o = \{(c_{r_1}, c_{r_1+1}, \cdots, c_{q_1}), (c_{r_2}, c_{r_2+1}, \cdots, c_{q_2}), \cdots, (c_{r_l}, c_{r_l+1}, \cdots, c_{q_l})\}$$

where $1 \leq q_i < r_{i+1} \leq k$ for $i \in \{1, 2, \cdots, l\}$. Consider the rest parts of the cycle in $G$:

$$r_G = \{(c_{q_1}, c_{q_1+1}, \cdots, c_{r_2}), (c_{q_2}, c_{q_2+1}, \cdots, c_{p_3}), \cdots, (c_{q_l}, \cdots, c_k, c_1, \cdots, c_{r_1})\}$$

Consider $c_{q_i}$ and $c_{r_{i+1}}$ for arbitrary $i$. Since they are included in $o$ and $o$ is a sequence, we have:

$$\text{Either } (c_{q_i}, c_{r_{i+1}}) \text{ or } (c_{r_{i+1}}, c_{q_i}) \text{ is contained in } o^+ \tag{13}$$

Recall that all the paths in $r_G$ belong to $G$, then the path $(c_{q_i}, c_{q_i+1}, \cdots, c_{r_{i+1}})$ belongs to $G$. Combined with the right-hand condition in (12), we have $(c_{q_i}, c_{r_{i+1}}) \notin o^+$. Then we have $(c_{q_i}, c_{r_{i+1}}) \in o^+$ by (13). This conclusion holds for all $i \in \{1, 2, \cdots, l\}$ (note that $r_{l+1} = r_1$). Then we have:

$$\{(c_{q_1}, c_{r_2}), (c_{q_2}, c_{r_3}), \cdots, (c_{q_l}, c_{r_1})\} \subset o^+ \tag{14}$$

Recall that paths in $r_o$ are contained in $o$, and we also have:

$$\{(c_{r_1}, c_{q_1}), (c_{r_2}, c_{q_2}), \cdots, (c_{r+l}, c_{q_l})\} \subset o^+ \tag{15}$$

Combining (14) and (15), we derive that a cycle $(c_{r_1}, c_{q_1}, c_{r_2}, \cdots, c_{q_l}, c_{r_1})$ is contained in $o^+$, which conflicts the premise that $o$ is acyclic.

($\implies$) Since $E(G) \subset E(G')$ and $G$ is DAG, we have that $G$ is DAG. Since the path $o$ is contained in $G'$, we have $X_i \rightsquigarrow X_j \in G$ for all $(i, j) \in o^+$. Then we have that no path $X_j \rightsquigarrow X_i$ exists in $G'$ as it will introduce a cycle. The proof is completed. $\qquad\square$

Now we consider the second statement:

**Lemma 2.** *For the set $\mathcal{P}(\mathcal{O}^-)$ of all maximal paths of $\mathcal{O}^-$, the union of the transitive closures of these paths is the transitive closure of $\mathcal{O}$: $\bigcup_{o \in \mathcal{P}(\mathcal{O}^-)} o^+ = \mathcal{O}^+$*

*Proof.* We have $o^+ \subseteq \mathcal{O}^+$ by that $o \subseteq \mathcal{O}^- \subseteq \mathcal{O}$. Therefore $\bigcup_{o \in \mathcal{P}(\mathcal{O}^-)} o^+ \subseteq \mathcal{O}^+$. Then we prove $\mathcal{O}^+ \subseteq \bigcup_{o \in \mathcal{P}(\mathcal{O}^-)} o^+$ by contradiction. Suppose that $\exists (i, j) \in \mathcal{O}^+$ such that $(i, j) \notin \bigcup_{o \in \mathcal{P}(\mathcal{O}^-)} o^+$. We have that $(i, j) \in (\mathcal{O}^-)^+$ by that $(\mathcal{O}^-)^+ = \mathcal{O}^+$. This indicates that a path from $i$ to $j$ exists in $\mathcal{O}^-$. Moreover, we have that this path does not belong to any maximal path of $\mathcal{O}^-$ by that $(i, j) \notin \bigcup_{o \in \mathcal{P}(\mathcal{O}^-)} o^+$. This introduces contradiction since any path in a graph at least belongs to one of its maximal path (we can extend any path to be a maximal path). $\square$

Now we derive the result of Theorem 1 by these results. Consider the result of Lemma 1:

$$G' \in \text{DAG for } E(G') = E(G) \cup o \iff G \in \text{DAG and } X_j \rightsquigarrow X_i \notin G \text{ for } (i, j) \in o^+$$

The left-hand condition equals $h(\mathcal{A}(W, o)) = 0$. Combining this with Equation (11), we have:

$$h'(W, \mathcal{O}) = 0 \iff G \in \text{DAG and } X_j \rightsquigarrow X_i \notin G \text{ for } (i, j) \in \cup_{o \in \mathcal{P}(\mathcal{O}^-)} o^+$$

With the result $\cup_{o \in \mathcal{P}(\mathcal{O}^-)} o^+ = \mathcal{O}^+$ from Lemma 3, we have:

$$h'(W, \mathcal{O}) = 0 \iff G \in \text{DAG and } X_j \rightsquigarrow X_i \notin G \text{ for } (i, j) \in \mathcal{O}^+$$

The path absence condition on the right-hand side is equivalent to adherence to $\mathcal{O}$. Hence, we complete the proof of Theorem 1.

### C.2 Proof of the rest statements

**Proposition 1.** *(Theorem 1 in [Wei et al., 2020]). The directed graph of a non-negative adjacency matrix $B$ is a DAG if and only if $h(B) = 0$ for any $h$ defined by (4).*

*Proof.* Recall the definition of $h(B)$:

$$h(B) = \text{Trace}(\sum_{i=1}^{d} c_i B^i), \ c_i > 0$$

This function can be understood by interpreting the diagonal elements of the matrix powers $B^i$, which represent the weighted $i$-length directed paths from a variable to itself, essentially, a cycle. A matrix $B$ represents a DAG if and only if there are no $i$-length cycles for any $i \in \mathbb{N}^+$, evidenced by $\text{Trace}(B^i) = 0$. Consequently, $\text{Trace}(B^i) = 0$ must hold for all $i \in \mathbb{N}^+$ if $B$ describes a DAG. The converse is supported by Lemma 3, which confirms that the absence of cycles ranging from 1-length to $d$-length, as characterized by Equation (4), sufficiently ensures the acyclicity of the graph. $\square$

**Lemma 3.** *Any cyclic graph with $d$ variables must contain cycle(s) with less than $d$ length.*

*Proof.* We prove this lemma by contradiction. Suppose that the cycle with the minimum length $l > d$ as $(i_1, i_2, \cdots, i_l, i_1)$. Since there are $d$ nodes, there exist at least two indexes in the cycle $i_q, i_p$ that refers to the same node, i.e., $i_q = i_p$. Suppose that that $q < p$, then the new cycle $(i_1, \cdots, i_{q-1}, i_{p+1}, \cdots, i_1$ by cutting the paths from $i_q$ to $i_p$ has a lower length. This conflicts with that the original cycle has the minimum length. $\square$

**Proposition 3.** *A graph $G$ is a DAG and satisfies total ordering $\pi$ if and only if edge $(u, v)$ does not exist in $G$ for all $(v, u) \in \pi^+$.*

*Proof.* Firstly, we formalize this statement according to Proposition 4 as follows:

$$(u, v) \notin G \text{ for } (v, u) \in \pi^+ \iff G(W) \in \text{DAG and } X_u \rightsquigarrow X_v \notin G \text{ for } (v, u) \in \pi^+$$

The necessity ($\Longleftarrow$) is straightforward by that $X_u \rightsquigarrow X_v \notin G \implies (u, v) \notin G$. For sufficiency ($\Longrightarrow$), we employ a proof by contradiction. Assume that there exists a path $(u_1, u_2, \ldots, u_k)$ in $G$

that violates $\pi$, such that $(u_k, u_1) \in \pi^+$. For $u_i$ and $u_{i+1}$, either $(u_i, u_{i+1}) \in \pi^+$ or $(u_{i+1}, u_i) \in \pi^+$ given that the order of any pairwise nodes in contained in $\pi^+$. Combined with the transitivity of orders, there must be at least one edge $(u_i, u_{i+1}) \in E(G)$ where $(u_{i+1}, u_i) \in \pi^+$. This contradicts the condition that $(u, v) \notin G$ for all $(v, u) \in \pi^+$. To demonstrate the absence of cycles (DAG constraint), consider setting $u_k = u_1$, which similarly leads to a contradiction under these conditions. $\square$

**Proposition 2.** *There exists at least one topological sort of DAG $G$ that satisfies the partial order set $\mathcal{O}$ if and only if, for any order $(i, j)$ in $\mathcal{O}^+$, $X_j$ is not an ancestor of $X_i$ in $G$.*

*Proof.* ($\Longrightarrow$) If there exists a topological sort that satisfies $\mathcal{O}$, then for any $(i, j) \in \mathcal{O}^+$, $X_j$ is not an ancestor of $X_i$ in $G$.

Suppose there exists a topological sort $T$ of $G$ that satisfies $\mathcal{O}$. For any $(i, j) \in \mathcal{O}^+$, $i$ must appear before $j$ in the topological sort $T$ because $\mathcal{O}^+$ represents a transitive closure of the order constraints. If $X_j$ were an ancestor of $X_i$ in $G$, there would be a path from $X_j$ to $X_i$. In a topological sort, for any edge $(u, v)$, $u$ must appear before $v$. Therefore, if $X_j$ were an ancestor of $X_i$, $X_j$ would appear before $X_i$ in the topological sort. However, this would contradict the fact that $i$ appears before $j$ in the topological sort $T$ (as $i$ is related to $j$ by $\mathcal{O}^+$). Thus, $X_j$ cannot be an ancestor of $X_i$ in $G$.

($\Longleftarrow$) If for any order $(i, j)$ in $\mathcal{O}^+$, $X_j$ is not an ancestor of $X_i$ in $G$, then there exists a topological sort of $G$ that satisfies $\mathcal{O}$.

Assume for any $(i, j) \in \mathcal{O}^+$, $X_j$ is not an ancestor of $X_i$ in $G$. This implies there is no directed path from $X_j$ to $X_i$ for any $(i, j) \in \mathcal{O}^+$). Construct a topological sort of $G$ using a standard topological sorting algorithm (such as Kahn's Algorithm or Depth-First Search). During the construction, ensure that for any $(i, j) \in \mathcal{O}$, $i$ appears before $j$ in the ordering. Since $\mathcal{O}^+$ does not create any cycles (because $X_j$ is not an ancestor of $X_i$), the ordering respects the partial order $\mathcal{O}$. The resulting topological sort $T$ satisfies the constraints of $\mathcal{O}$ by construction. $\square$

**Proposition 4.** *No directed path $X_i \rightsquigarrow X_j$ exists in $G(W)$ if and only if $\left( \sum_{l=1}^{d} (W \circ W)^l \right)_{i,j} = 0$.*

*Proof.* $\sum_{l=1}^{d} (W \circ W)^l$ is an instance of the acyclicity function defined in Equation (4). By the proof of Proposition 1, setting the matrix entry $(i, j)$ to zero effectively blocks all paths of lengths ranging from 1 to $d$ from $X_i$ to $X_j$, fulfilling the necessary condition for $X_i \rightsquigarrow X_j \notin G(W)$. This condition is also deemed sufficient due to Lemma 4. $\square$

**Lemma 4.** *If a directed path $X_i \rightsquigarrow X_j$ exists in a graph with $d$ nodes, then a directed path $X_i \rightsquigarrow X_j$ of length less than $d$ must also exist.*

The proof of Lemma 4 is similar to that of lemma 3.

## Appendix D  Implementation of the Proposed Method

This section presents the psudocodes of the implementations of the proposed method.

---

**Algorithm 1** Partial Order Constraint-based Differentiable Structure Learning

---

**Require:** Observational data $D \in \mathbb{R}^{m \times d}$; A set of partial orders $\mathcal{O}$.
**Ensure:** A DAG $G$.
 1: $W \leftarrow$ Solve $\min_W \mathcal{F}(W, D)$ subject to $h'(W, \mathcal{O}) = 0$ by a backbone algorithm $\triangleright$ $h'$ is defined by Algorithm 2.
 2: Construct $G$ by adding edges $(i, j)$ where $|W_{i,j}| > \gamma$ $\triangleright \gamma > 0$ is the threshold value, which is set to 0.3 in experiments.
 3: **return** $G$

---

---
**Algorithm 2** Augmented Acyclicity Characterization Function $h' : \mathbb{R}^{d \times d} \to \mathbb{R}$
---
**Require:** A weighted matrix $W \in \mathbb{R}^{d \times d}$; A set of partial orders $\mathcal{O}$; An acyclicity characterization function $h : \mathbb{R}^{d \times d} \to \mathbb{R}$.
1: $\mathcal{O}^- \leftarrow$ Derive the transitive reduction of $\mathcal{O}$ by Algorithm 4
2: $G \leftarrow$ Construct a graph by adding edges $(i, j)$ where $(i, j) \in \mathcal{O}^-$
3: $result \leftarrow 0$
4: $\mathcal{P} \leftarrow$ Find all maximal paths in $G$ by Algorithm 3
5: **for** each path $o$ in $\mathcal{P}$ **do**
6: $\quad W' \leftarrow$ Replace the $(i, j)$th element of $W$ with $\tau$ for each edge $(i, j) \in o$ $\qquad \triangleright \tau > 0$ is a hyper-parameter that controls the strength of the order constraints.
7: $\quad result \leftarrow result + h(W')$
8: **end for return** $result$
---

---
**Algorithm 3** Find All Maximal Paths in a DAG
---
**Require:** A directed acyclic graph $G = (V, E)$
**Ensure:** A set of all maximal paths in $G$
1: **function** FINDMAXIMALPATHS($G$)
2: $\quad$ Initialize an empty list $all\_paths$
3: $\quad$ Identify all nodes in $G$ with no incoming edges as $start\_nodes$
4: $\quad$ **for** each node $u$ in $start\_nodes$ **do**
5: $\quad\quad$ Call DFS($u$, [])
6: $\quad$ **end for**
7: $\quad$ **return** $all\_paths$
8: **end function**
9: **function** DFS($node$, $path$)
10: $\quad$ Append $node$ to $path$
11: $\quad$ Initialize $extensions\_found$ to $False$
12: $\quad$ **for** each $v$ such that there is an edge from $node$ to $v$ **do**
13: $\quad\quad$ Set $extensions\_found$ to $True$
14: $\quad\quad$ Call DFS($v$, $path$)
15: $\quad$ **end for**
16: $\quad$ **if** not $extensions\_found$ **then**
17: $\quad\quad$ Add $path$ to $all\_paths$ $\qquad\qquad\qquad\qquad\qquad \triangleright$ Path is maximal
18: $\quad$ **end if**
19: $\quad$ Remove $node$ from $path$ $\qquad\qquad\qquad\qquad\qquad\qquad \triangleright$ Backtrack
20: **end function**
---

---
**Algorithm 4** Derive Transitive Reduction of a Relation
---
**Require:** A set of ordered pairs $\mathcal{O}$ representing a relation.
**Ensure:** Transitive reduction of $\mathcal{O}$.
1: Compute $R$, the transitive closure of $\mathcal{O}$
2: Initialize $T \leftarrow \mathcal{O}$
3: **for** each pair $(i, j) \in \mathcal{O}$ **do**
4: $\quad$ **for** each pair $(k, l) \in \mathcal{O}$ **do**
5: $\quad\quad$ **if** $i \neq k$ and $j \neq l$ **then**
6: $\quad\quad\quad$ **if** $(i, k) \in R$ and $(k, j) \in R$ **then**
7: $\quad\quad\quad\quad$ Remove $(i, j)$ from $T$
8: $\quad\quad\quad$ **end if**
9: $\quad\quad$ **end if**
10: $\quad$ **end for**
11: **end for**
12: **return** $T$
---

# Appendix E    Complete Experimental Results

This section presents the complete experimental results and detailed experimental settings. The hyper-parameters of algorithms include: the threshold $\gamma = 0.3$ for the existence of an edge, the weight $\lambda_1 = 0.03$ for L1 regularization term, the weight $\lambda_2 = 0.01$ for L2 regularization term, the weight $\tau = 1$ in the proposed augmented acyclicity term, the maximum $\rho$ value $\rho_{\max} = 10^{16}$ for augment Lagrangian method. The NOTEARS-MLP uses an MLP with an input dimension $d$, a hidden layer of size $d \times 10$, and an output layer of size $d$, where $d$ is the number of variables. The evaluation settings are detailed in the caption of each figure.

The evaluation metrics used in the study include the following: Structural Hamming Distance (SHD), which counts the number of edges that differ between the recovered structure and the ground truth; True Positive Rate (TPR), defined as $TP/(TP + FN)$, where $TP$ is the number of correctly recovered edges and $FN$ is the count of missing or reversed edges; False Discovery Rate (FDR), calculated as $FP/(FP + TP)$, where $FP$ represents the number of extra edges; and the F1-score, computed as $2 \cdot P/(P + R)$, with $R = \text{TPR}$ and $P = 1 - \text{FDR}$.

The runtime of the algorithm refers to the time taken to generate the final result, specifically the time required for the acyclicity term $h(\cdot)$ to reach a sufficiently small value (set to $10^{-8}$ in the experiments).

We first report two supplementary results regarding 1) the comparison of the pure path absence-based implementation partial order constraints as shown in Equation (8) and 2) the selection of hyper-parameter $\tau$ in Equation (9c). Following this, we report the supplementary results to those in the main texts.

Table 2: Comparison of path absence- and augmented acyclicity-based partial ordering constraints for structure learning in terms of run time $t$ (s) and output quality F1-score.

| Prior | Metric | d=20 | d=30 | d=40 | d=50 |
|-------|--------|------|------|------|------|
| Na | *t / F1* | *18.3 / 0.47* | *35.1 / 0.51* | *44.7 / 0.54* | *71.6 / 0.49* |
| 0.1d | t (s) | **19.7** / 107.3 | **41.1** / 170.9 | **65.8** / 316.5 | **117.5** / 6523.0 |
|      | F1 | 0.42 / 0.39 | 0.49 / 0.52 | 0.56 / 0.54 | 0.48 / 0.52 |
| 0.5d | t (s) | **26.2** / 328.3 | **21.3** / 221.1 | **41.9** / 271.8 | **106.7** / 4124.5 |
|      | F1 | 0.50 / 0.53 | 0.60 / 0.62 | 0.59 / 0.57 | 0.52 / 0.52 |
| d | t (s) | **4.2** / 140.0 | **7.2** / 56.5 | **12.1** / 83.0 | **41.9** / 1475.3 |
|   | F1 | 0.71 / 0.71 | 0.75 / 0.75 | 0.72 / 0.72 | 0.70 / 0.70 |

## E.1    Comparison of Path Absence- and Augmented Acyclicity-based Partial Order Constraints

We implemented Equation (8), which applies straightforward path absence constraints to enforce partial order, and compared its performance with our augmented acyclicity-based method. The results, averaged over 3 repetitions, in terms of runtime and F1-score using NOTEARS are shown in Table 2. The experimental settings include: ER-4 structure, linear data with Gaussian noise, a sample size of 40, and a single-chained partial order with nodes $[0.1d, 0.5d, d]$. Results are presented as Ours/Equation 8, with Prior Na representing the baseline (NOTEARS without prior constraints).

The results demonstrate that the method in Equation (8) achieves comparable output quality to our main approach, showing that it effectively represents partial order constraints. However, as expected, Equation (8) exhibits significantly slower runtime. Interestingly, the most significant slowdown occurs when fewer variables are included in the ordering than the total variable set. This may be due to the method's disjoint integration with the acyclicity term, possibly causing the partial order term to take longer to reconcile with the acyclicity constraint when the prior information is insufficient.

## E.2    Analysis of Selection of Virtual Edge Weight $\tau$

The hyper-parameter $\tau > 0$ serves as the uniform weight for edges in the set $\mathcal{P}(\mathcal{O}^-)$, which are added solely to the augmented acyclicity term $h'(W, \mathcal{O})$ and not to the data approximation term

Table 3: Variance of different properties in partial order-based structure learning with varying $\tau$ values.

| $\tau$ | 0 | 0.1 | 0.2 | 0.3 | 0.4 | 0.5 | 1 | 2 | 3 | 4 | 5 | 6 | 7 | 8 |
|---|---|---|---|---|---|---|---|---|---|---|---|---|---|---|
| $h$ | 4.E-01 | 1.E-04 | 2.E-01 | 2.E-06 | 4.E-07 | 1.E-09 | 4.E-09 | 1.E-10 | 5.E-11 | 6.E-09 | 3.E-09 | 7.E-05 | 1.E-12 | 0.E+00 |
| $h'$ | 6.E-09 | 2.E-05 | 6.E-04 | 2.E-08 | 2.E-08 | 3.E-08 | 6.E-09 | 5.E-09 | 4.E-09 | 2.E-10 | 8.E-08 | 4.E+00 | 1.E-02 | 5.E-02 |
| $\mathcal{F}$ | 9.6 | 5.E+03 | 4.E+03 | **12.9** | **12.7** | **16.9** | **11.7** | **11.7** | **11.5** | **11.7** | 8.E+02 | 2.E+04 | 3.E+04 | 3.E+04 |
| DAG? | ✗ | ✗ | ✗ | ✓ | ✓ | ✓ | ✓ | ✓ | ✓ | ✓ | ✓ | ✓ | ✓ | ✓ |
| @Edge | 53 | 31 | 55 | 61 | 62 | 69 | 71 | 70 | 66 | 67 | 33 | 1 | 0 | 0 |
| F1 | 0.27 | 0.27 | 0.36 | 0.61 | 0.65 | 0.56 | 0.68 | 0.69 | 0.7 | 0.69 | 0.23 | 0 | 0 | 0 |

$\mathcal{F}(W)$. These edges, termed "virtual" edges, may not exist in the actual graph but are included in the "acy-graph" used to enforce acyclicity constraints. The following analysis highlights the theoretical impact of different values of $\tau$.

Small $\tau$: When $\tau$ is too small, it fails to properly reflect the presence of virtual edges in the acy-graph, leading to inadequate enforcement of prior information. This can result in reversed paths not being properly excluded, weakening the acyclicity constraint and potentially allowing cycles to form. Essentially, a very small $\tau$ removes the effect of virtual edges in $\mathcal{P}(\mathcal{O}^-)$, diminishing the strength of the acyclicity constraint and compromising the learning process.

Large $\tau$: On the other hand, if $\tau$ is too large, it can disrupt numerical stability by overshadowing small weights that signify absent edges. For example, consider a forbidden cycle $p = W_{1,2}W_{2,3}W_{3,1}$, where $W_{1,2} = \tau$ is set to a large value, while $W_{2,3} = r^-$ (indicating edge absence). This configuration results in $\nabla_{W_{3,1}} p = \tau \times r^-$, making the influence of $W_{1,2}$ on $W_{3,1}$ large enough to enforce the absence of $W_{3,1}$, even though the cycle should be considered broken due to the absence of $W_{2,3}$. This misinterpretation can lead to the erroneous removal of edges, potentially producing an empty graph.

We conducted experiments with varying values of $\tau$ and observed the real acyclicity loss $h$, augmented acyclicity loss $h'$, data approximation loss $\mathcal{F}$, and output metrics such as DAG condition, edge count, and F1 score. Results for an ER-4 graph with Gaussian noise model, node count $d = 20$, sample size $n = 40$, and edge threshold $\gamma = 0.1$ support this theoretical analysis and are presented in Table 3.

## E.3 Supplementary Results

This section presents supplementary results to the main text, providing additional insights across complete datasets, various data settings, and methods used. The figures included, Figures 4-8, expand on the analysis, offering a more comprehensive understanding of the experimental outcomes.

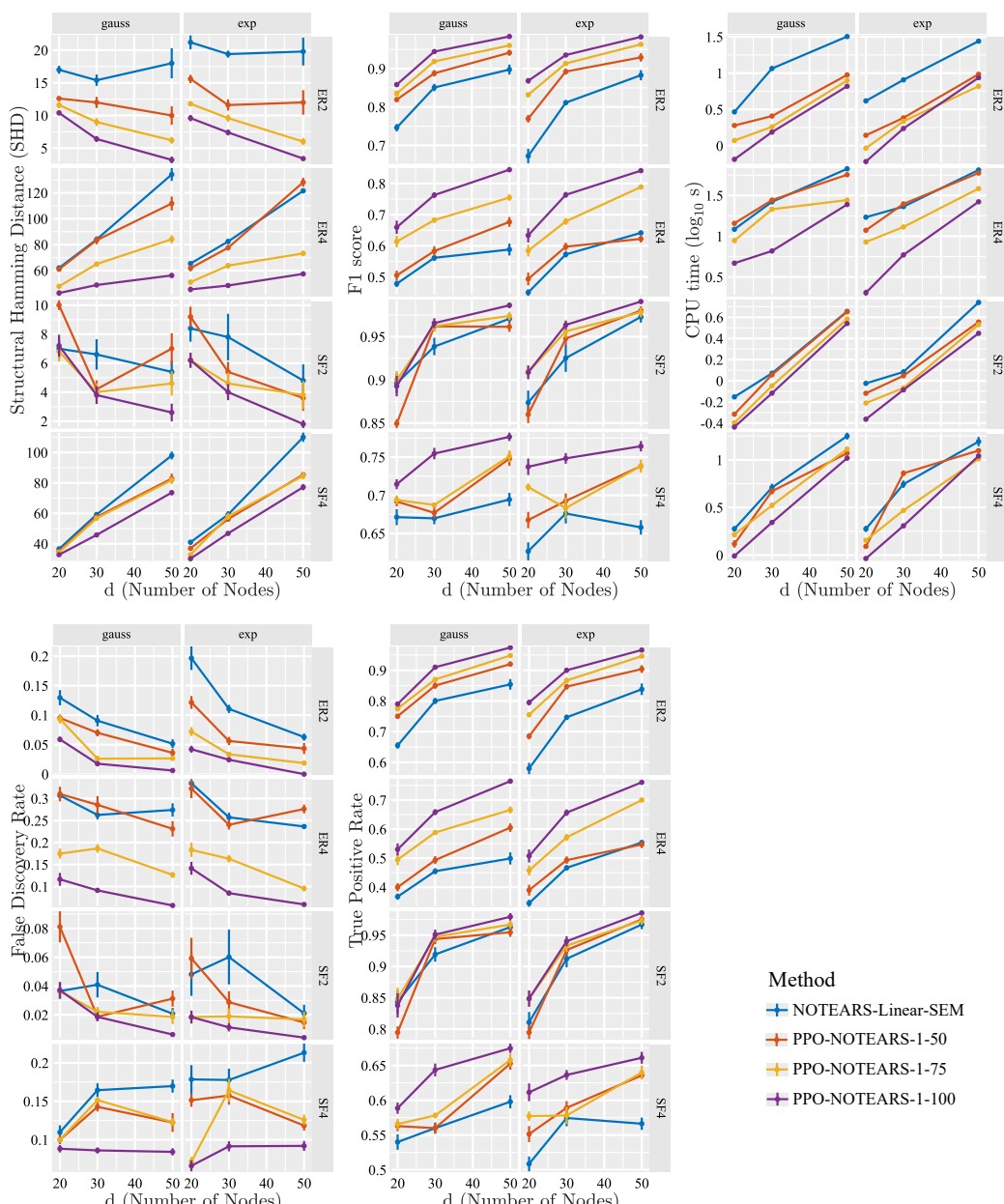

Figure 4: Results of single-chained ordering for NOTEARS with linear SEM. Structural discovery in terms of SHD (lower is better), F1-score (higher is better), FDR (lower is better), TPR (higher is better) and CPU time ($\log_{10}$ s). Rows: graph types. {ER,SF}-$k$ represents {Erdös-Rényi, scale-free} graphs with $kd$ expected edges. Columns: noise types of SEM. Error bars represent standard errors over 5 simulations. Method: PPO-NOTEARS-1-$p$ represents our method with the ordering of a single chain with size $m = p\%d$.

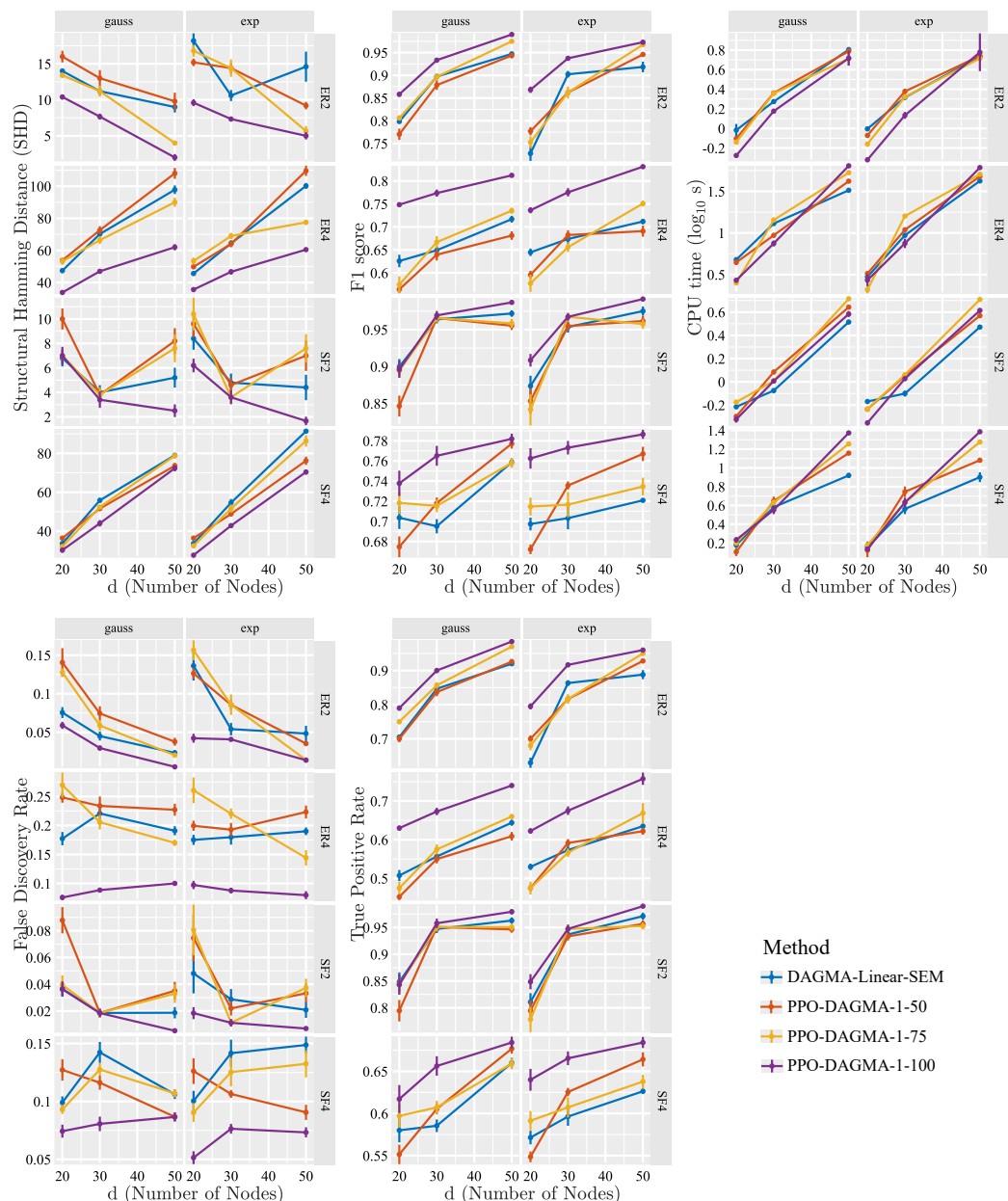

Figure 5: Results of single-chained ordering for DAGMA. Structural discovery in terms of SHD (lower is better), F1-score (higher is better), FDR (lower is better), TPR (higher is better) and CPU time ($\log_{10}$ s). Rows: graph types. {ER,SF}-$k$ represents {Erdös-Rényi, scale-free} graphs with $kd$ expected edges. Columns: noise types of SEM. Error bars represent standard errors over 5 simulations. Method: PPO-DAGMA-1-$p$ represents our method with the ordering of a single chain with size $m = p\%d$.

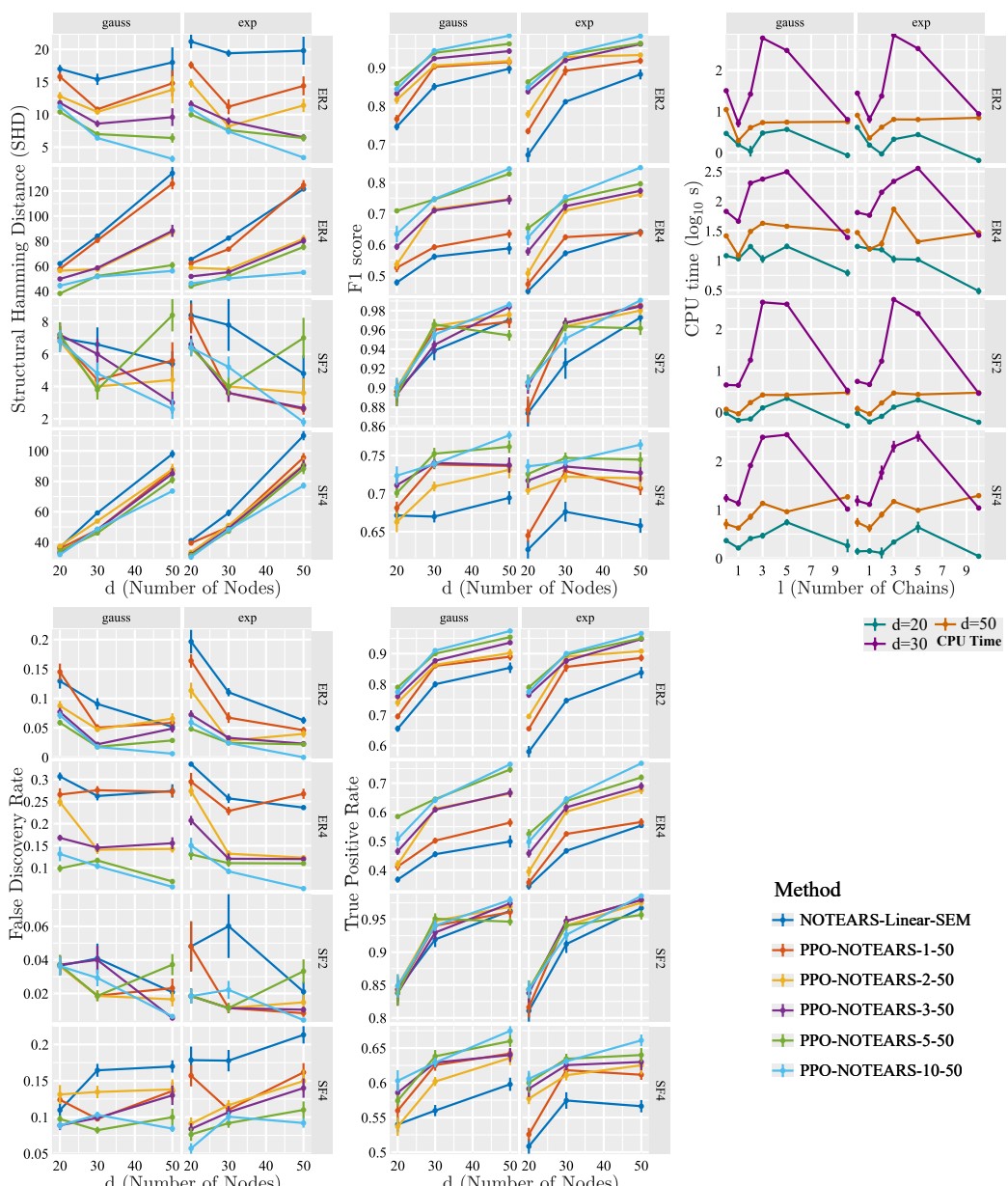

Figure 6: Results of multi-chained orderings for NOTEARS with linear SEM. Structural discovery in terms of SHD (lower is better), F1-score (higher is better), FDR (lower is better), TPR (higher is better) and CPU time ($\log_{10}$ s). Rows: graph types. {ER,SF}-$k$ represents {Erdös-Rényi, scale-free} graphs with $kd$ expected edges. Columns: noise types of SEM. Error bars represent standard errors over 5 simulations. Method: PPO-NOTEARS-$l$-50 represents our method using the ordering where the chain number is $l$ and the chain size is $m = 0.5d$.

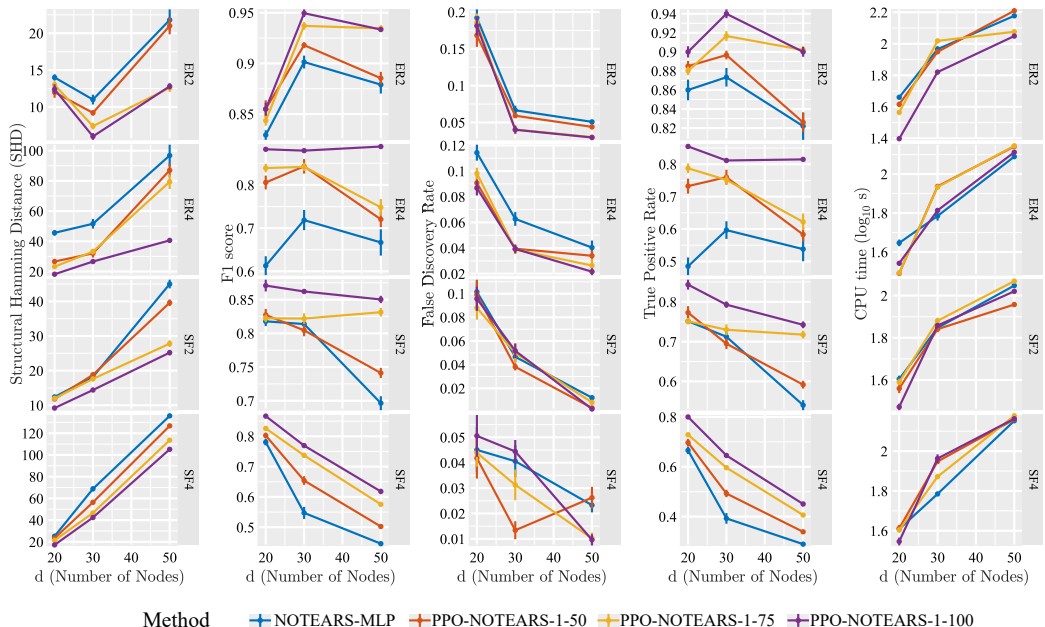

Figure 7: Results of single-chained Ordering for NOTEARS-MLP. Structural discovery in terms of SHD (lower is better), F1-score (higher is better), FDR (lower is better), TPR (higher is better) and CPU time ($\log_{10}$ s). Rows: graph types. {ER,SF}-$k$ represents {Erdös-Rényi, scale-free} graphs with $kd$ expected edges. Error bars represent standard errors over 5 simulations. Method: PPO-NOTEARS-1-$p$ represents our method with the ordering of a single chain with size $m = p\%d$.

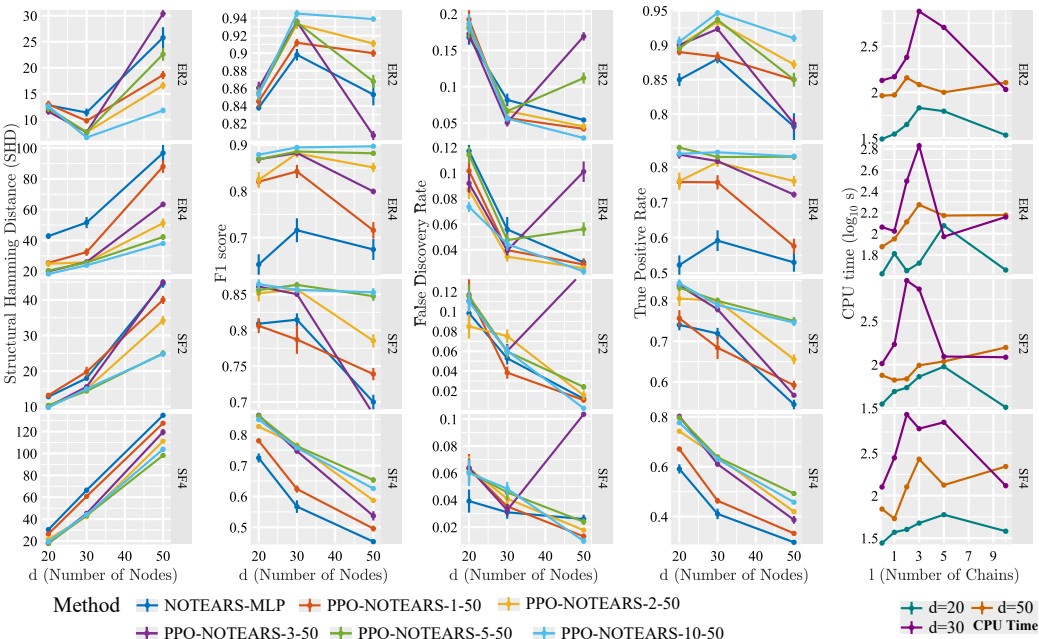

Figure 8: Results of multi-chained orderings for NOTEARS-MLP. Structural discovery in terms of SHD (lower is better), F1-score (higher is better), FDR (lower is better), TPR (higher is better) and CPU time ($\log_{10}$ s). Rows: graph types. {ER,SF}-$k$ represents {Erdös-Rényi, scale-free} graphs with $kd$ expected edges. Error bars represent standard errors over 5 simulations. Method: PPO-NOTEARS-$l$-50 represents our method using the ordering where the chain number is $l$ and the chain size is $m = 0.5d$.

