# OpenReview forum: "Differentiable Structure Learning with Partial Orders"
_NeurIPS.cc/2024/Conference — NeurIPS 2024 poster_

### Official Review · Reviewer_FFcG · 2024-07-11

**Soundness:** 3
**Presentation:** 3
**Contribution:** 3
**Rating:** 7
**Confidence:** 4

**Summary:**

This paper studies the problem of learning DAGs from observational data with incorporating prior knowledge represented as partial order constraints by extending from existing continuous DAG learning methods such as NOTEARS and DAGMA. The paper first presents a method that shuts down the corresponding cells of the adjacency matrix of the DAG according to the permutations of the partial orders and then proposes an augmented acyclicity-based method with improved efficiency. The paper conducts experiments on simulated datasets and a real-world dataset. In addition, the paper also provides comprehensive theoretical motivations and analysis to the research problem and the proposed method.

**Strengths:**

1. The problem studied in the paper is novel and has significance in practice. To my knowledge, it is a new (sub-)problem of DAG discovery from observational data, which has not been studied before.

2.  The proposed method is well motivated both conceptually and theoretically. The notations and definitions are rigorously defined.

3. The experiments are comprehensive and support the claims of the paper.

**Weaknesses:**

Not all my comments below are weaknesses and some of them are questions.

1. In Eq 8a, a (relatively) straightforward method is introduced, which is argued to be less efficient, motivating the development of the augmented acyclicity-based method. Is the straightforward method implementable? If so, it would be better to compare it with augmented acyclicity-based method in terms of the performance and efficiency.

2. I find it hard to fully understand Eq 9c and 9d. What's the definition of $\mathcal{A}(W, o)$ and why does it have the formulation in 9c? What's $W_{o,i,j}$? Does it mean $W$ is a three dimensional tensor?

3. To learn $W$, one needs the algorithm to be differentiable in terms of $W$. For the method in Eq 8a, one might need to backpropagate gradients through permutations. If so, how can we do this without continuous relaxation? For the augmented acyclicity-based method, it seems that one needs to conduct a few algorithms (Algorithm 2, 3, 4). Are the operations in these algorithms differentiable in terms of $W$? More discussions are needed.

4. The proposed method introduces a new hyperparameter $\gamma$. How to determine $\gamma$ in practice as there is no ground truth to do validation?

5. In the experiments of real-word data, Linear NOTEARS is used as the baseline. How do we know the (non)linearity in the real data?

**Questions:**

Please see my comments above.

**Limitations:**

The authors adequately addressed the limitations

---

> ### Author Rebuttal · Authors · 2024-08-02
>
> Thank you for your detailed review and insightful questions.  Here are our responses:
>
> 1. **Comparison with Eq. 8a:**
>
>    We have implemented Eq. 8 and compared it with our augmented acyclicity-based method. The results (average of 3 repetitions) using NOTEARS are reported here. Settings are: ER-4, linear data with gauss noise, sample size is 40, single-chained partial order with nodes $[0.1d,0.5d,d]$. **Results are reported as Ours/Eq. 8. Prior Na denotes the baseline (NOTEARS without prior).**
>
>
> | Prior  | Metric | d=20           | d=30           | d=40           | d=50             |
> | ------ | ------ | -------------- | -------------- | -------------- | ---------------- |
> | Na     | *t*/F1 | *18.3*/.47     | *35.1*/.51     | *44.7*/.54     | *71.6*/.49       |
> | $0.1d$ | t (s)  | **19.7**/107.3 | **41.1**/170.9 | **65.8**/316.5 | **117.5**/6523.0 |
> |        | F1     | .42/.39        | .49/.52        | .56/.54        | .48/.52          |
> | $0.5d$ | t (s)  | **26.2**/328.3 | **21.3**/221.1 | **41.9**/271.8 | **106.7**/4124.5 |
> |        | F1     | .50/.53        | .60/.62        | .59/.57        | .52/.52          |
> | $d$    | t (s)  | **4.2**/140.0  | **7.2**/56.5   | **12.1**/83.0  | **41.9**/1475.3  |
> |        | F1     | .71/.71        | .75/.75        | .72/.72        | .70/.70          |
>
>
>    - The results show that the method in Eq. 8 produces comparable output quality with our main approach, indicating that it can correctly represent the partial order constraint.
>    - As expected, Eq. 8 has significantly slower run time. Interestingly, the worst efficiency occurs with fewer variables than the total variables in the ordering, possibly due to its disjoint manner with the acyclicity term. This might (not very clear) cause the partial order term to take more time to balance with the acyclicity term when the prior is not sufficient.
>
> 2. **Explanation on $\mathcal{A}(W,o):$**
>
>    $\mathcal{A}(W, o)$ sets the edges in the path $o$ to a uniform weight to indicate their existence.
>
>    - $W_{o,i,j}$ is the $(i,j)$th element of $W_o$ (as illustrated in Notations), a mask matrix where edges in $o$ are 1 and others are 0.
>    - For $\mathcal{A}(W, o)$ in Eq. 9c, we remove the original weights in $W$ for the edges in $o$ by $W - W \circ W_o$, and then add the edges in $o$ with a uniform weight $\tau$ by $W - W \circ W_o + \tau W_o$. This operation 1) avoids the sum of $\tau>0$ to negative weights that can lead to *accidental removal* of edges, and 2) ensures numerical stability by uniforming the weights.
>
> 3. **Differentiability of Algorithms:**
>
>    Thanks for your insightful question. Indeed, transforming a discrete constraint to a continuous one typically requires some continuous relaxation. The differentiable nature of our constraint terms is illustrated as follows.
>
>    - In Eq. 8a, the prior information of partial orders is transformed into the sum of multiple path absence terms for each pair in the transitive closure $\mathcal{O}^+$ of partial orders. Each path absence term is naturally continuous because the structural negative constraint aims to set corresponding structural parameters to strictly 0 without requiring additional relaxation. For example, to forbid an edge $(x_i, x_j)$, we add a penalty $\lambda|W_{i,j}|$ to push it towards 0, similar for path absence. Conversely, for edge existence, a threshold is typically used to confirm its presence, with a possible penalty being $\lambda \text{ReLU}(\text{thr}-|W_{i,j}|)$.
>
>    - For the augmented acyclicity method, Algorithms 3 and 4 preprocess $\mathcal{O}$ to derive $W_o$, and Algorithm 2 calculates the augmented acyclicity term with respect to the continuous function presented in Eqs. 9b and 9c, thus ensuring the differentiability.
>
> 4. **Selection on Hyper-Parameter**
>
>    We assume you are referring to $\tau$ in Eq. 9c, a new hyperparameter introduced by our method. All other hyperparameters remain the same as in NOTEARS. Here is how to select $\tau$ without ground truth:
>
>    - Start with a relatively small value and incrementally increase it, observing the **number of recovered edges** by the algorithm. Choose $\tau$ values that maintain a stable number of recovered edges.
>    - Due to text limitations, please refer to our reply to **Weakness 5 to Reviewer 99Ub** for an in-depth analysis. The optimization process is sensitive to $\tau$; too small a $\tau$ can lead to a weaker acyclicity constraint and result in cycles, while too large a $\tau$ can erroneously enforce the absence of edges. Fortunately, the stable range of $\tau$ is relatively wide, as indicated in the experiment results and related discussions in the reply to Reviewer 99Ub.
>
> 5. **Nonlinear Results on Sach**
>
>    - Indeed, we do not know the (non)linearity of data in practice. Hence, we add an experiment using NOTEARS-MLP to fit possible nonlinear functions in the Sach-853 dataset. The MLP layers are $11, 11 \times 10, 1$, with L1-reg weight of 0.1 and L2-reg weight of 0.03. Results:
>
>    |             | SHD  | F1   | TPR  |
>    | ----------- | ---- | ---- | ---- |
>    | NOTEARS-MLP | 39   | 0.21 | 0.35 |
>    | PPO-1-6     | 40   | 0.29 | 0.53 |
>    | PPO-1-8     | 26   | 0.29 | 0.35 |
>    | PPO-1-11    | 17   | 0.44 | 0.41 |
>
>    - The nonlinear baseline without prior performs much worse than the linear method, likely due to noise or unobserved influences in real-world data. However, integrating more partial orders improves SHD and F1 scores, leading to a structure closer to the ground truth. This indicates the effectiveness of using partial orders to improve nonlinear models.
>
> Thank you again for your insightful feedback, which has been invaluable in refining our paper. Please let us know if any further clarifications or additional information are needed.

---

> > ### Comment · Reviewer_FFcG · 2024-08-14
> >
> > Thanks for the detailed response. I am happy to keep my original positive score of the paper.

---

### Official Review · Reviewer_SXfo · 2024-07-11

**Soundness:** 2
**Presentation:** 3
**Contribution:** 2
**Rating:** 7
**Confidence:** 3

**Summary:**

This paper contributes interesting new theoretical results for the field of differentiable graph structure learning and showcases how to practically exploit those results for improved structure learning. Differentiable structure learning converts the combinatorial optimisation problem of finding the correct graph structure of a problem into an optimisation problem by parametrising an adjacency matrix according to the structural equation model (SEM). Existing methods focused on enforcing DAG structures on these parametrised matrices such that DAG graphs can be learned. This paper instead focuses on a new type of constraints, trying to enforce that the learned graphs adhere to a given set of orders on the underlying variables. It is first proven (Theorem 3) that adherence to a set of orders $\mathcal{O}$ is equivalent to a constraint comprised of $|\mathcal{O}^+|$ terms, which is deemed computationally infeasible. Then the paper continues to prove its main result, showing that adherence to the set of orders $\mathcal{O}$ is also equivalent to a constraint over maximal paths in the transitive *reduction* $\mathcal{O}^-$ of $\mathcal{O}$. Finally, this theoretical result is implemented such that it can take any existing SEM-based structure learning algorithm and augment it with given orders. The increase in performance is shown on a number of synthetic and more realistic structure learning benchmarks when comparing with and without the exploitation of given orders.

**Strengths:**

1. The structure of the paper is very clear and hence reads well. As someone who is not deeply familiar with the details of differentiable structure learning, the incremental and logical build-up of the preliminaries is also appreciated. In general, the authors put significant effort into making the paper self-contained, improving readability further.

2. The paper is very formal in its notation and methodology, without going overboard on mathematical notation or jargon. Again, this allows for someone not intimately familiar with the topic to more easily follow the proofs and reasoning of the paper.

3. The idea of the paper is certainly interesting and novel, especially considering the reference that incorporating orders can significantly reduce the search space of possible structures and the hardness of structure learning [1]. The overall idea of trying to incorporate prior knowledge of any form also ties in well with the rising popularity of neurosymbolic methods [2, 3].

4. The authors clearly also put in a lot of effort to ensure a reader can follow the theoretical arguments on a high level throughout the text without getting lost in mathematical intricacies during reading. The many intermediate comments and examples are very helpful to keep the story focused, which is never easy when the road to a general result requires multiple intermediate results.

5. The empirical evidence is very convincing. I thank the authors for including aggregates and variability metrics over multiple runs, which is sadly not a given anymore.

[1] Teyssier, M., & Koller, D. (2005, July). Ordering-based search: a simple and effective algorithm for learning Bayesian networks. In Proceedings of the Twenty-First Conference on Uncertainty in Artificial Intelligence (pp. 584-590).

[2] Garcez, A. D. A., & Lamb, L. C. (2023). Neurosymbolic AI: The 3 rd wave. Artificial Intelligence Review, 56(11), 12387-12406.

[3] Marra, G., Dumančić, S., Manhaeve, R., & De Raedt, L. (2024). From statistical relational to neurosymbolic artificial intelligence: A survey. Artificial Intelligence, 104062.

**Weaknesses:**

While I overall did enjoy reading the paper, I do have a number of questions and concerns:

1. The paper does not clearly distinguish its own theoretical contributions from previously known results. While I appreciate all lemmata and theorems are proven either in the main body or in the appendix, I would like to see clear statements of which results were already known, as some seem to relate to the provided references. In particular, I suspect Theorems 1, 2, and 3 were already proven before, though I am unsure about 3. Same for lemma 1. If some or all of these results are indeed novel, then explicitly stating they are can only further increase the impact of the paper.

2. Theorem 2 is a nice result showing how knowing a total ordering can indeed considerably reduce the complexity of a DAG structure learning task. However, it is not used in the rest of the paper. If Theorem 2 is not a new result, I would remove it to improve the flow of the paper and give more space for further clarifications.

3. I am not sure if I can agree with Remarks 2 and 6, where it is stated that the result of Theorem 3 does not lead to a computationally feasible solution while that of Theorem 4 does. It seems that using Equation 8b introduces a number of terms that is quadratic in the number of variables, as it is bounded by the number of possible pairs over those variables. While I can see that using Equation 9b is certainly more efficient for a single sequential ordering, the picture is less clear when $\mathcal{O}$ consists of many (sequential) orderings. In general, the trade-off between orderings in $\mathcal{O}^+$ or maximal paths in $\mathcal{O}^-$ does not seem clear, especially since no bounds on $|\mathcal{P}(\mathcal{O}^-)|$ are given.

4. While I generally agree with the provided proofs of all statements, there are a couple of places where I do have concerns about the validity of the results (see precise questions below). Importantly, I am unsure whether the exact statement of the main Theorem 4 holds, leading me to maintain a lower score for now until my questions are clarified.

5. The experimental section certainly considers enough datasets and data-generating configurations, but it does seem to miss one important comparison. I would have liked to see an experimental confirmation of the trade-off between using Equation 8b compared to Equation 9b, for both run time and general performance. Especially since the complexity of 8b is used as a motivation to discard it in favour of 9b. Moreover, some of the metrics could be better explained, even if only in the appendix. For example, I am not familiar with the details of the structural Hamming distance (SHD) or False Discovery Rate (FDR) and the notion of run time is also ambiguous.

**Questions:**

1. Can you clarify which theoretical statements are novel by specifying if Theorems 1, 2, and 3, and Lemma 1 were already known?

2. Theoretical clarifications:
   + I believe the proof of Lemma 2, specifically the necessity part, is not fully complete. The second case (line 543), where some edges of the cycle are contained in $o$, implicitly seems to assume that there are at least 2 disjoint paths in the cycle with edges in $o$, i.e. $|r_o| = l \geq 2$ because it is written that $q_i  < r_{i + 1}$. If $l = 1$, then $r_2$ does not exist and this inequality does not make sense. Hence, the case where $l = 1$ seems missing, although I think that a shorter version of a similar argument could apply.
   + I do not see how the argument in lines 567-569 follows from Lemma 3 together with Equation 16. Concretely, Lemma 3 **only** seems to say that simultaneous adherence to $o^+$ for all $o \in \mathcal{P}(\mathcal{O}^-)$ is equivalent to adherence to $\mathcal{O}^+$, **not** that adherence to $o$ for all $o \in \mathcal{P}(\mathcal{O}^-)$ is equivalent to adherence to $\mathcal{O}$. This confusion might be due to the ambiguity of notation as described in Remark 3, but it is important to validate Theorem 4. From Remark 3, it seems that you do not consider a sequential ordering, such as $o$, to be a partial ordering as it is not transitive (line 199). However, then $o$ and $o^+$ are not the same, implying that adherence to $o^+$ is stronger than adhering to $o$. Similarly, it is unclear whether $\mathcal{O}^+ = \mathcal{O}$. **This confusion can have strong implications on the impact of the paper and is my primary concern. If the reasoning in lines 567-569 is invalid, the proposed objective in Equation 9b only enforces a weaker version of ordering adherence than the one enforced by Equation 8b.**

3. Experimental clarification: What is the precise meaning of "run time" here? Is it the time required to learn the provided solutions? If so, what is the stopping condition for the algorithm?

4. Can you elaborate on the computational trade-off between using Equation 8b compared to Equation 9b? Do you believe there are cases where this trade-off is negligable or even in favour of Equation 8b?

5. Example 1 is not clear to me as it discusses the effect of removing **step 1** from the overall algorithm. However, it seems to be that **step 2** does not make sense without  **step 1**. So to be precise, why does $h'$ degenerate into $h(\mathcal{A}(W, \mathcal{O}^-)$ when removing **step 1**?

I want to stress that I am very open to significantly increase my overall score (and my scores of soundness and contribution) to a full accept if the authors can address my questions. In particular, the seeming uncertainty around the validity of Theorem 4 is my main reason for borderline rejection because of its potential impact. If Theorem 4 does end up being weaker than stated, I believe an empirical comparison between Equation 8b (which surely enforces strict partial order adherence) and Equation 9b is warranted.

**Limitations:**

The limitations are very well described in the appendix. I would only have like for it to be (partly) present somewhere in the main body of the paper, especially how any optimisation-based paradigm can not fully guarantee constraint satisfaction.

---

> ### Author Rebuttal · Authors · 2024-07-31
>
> Thank you for your detailed review and invaluable questions. We begin with addressing your major question on the proof part in the paper (point 2 in the Question section). Then, we will respond to your other questions point-by-point.
>
> **Regarding Eq. 8 and the comparison results:** We have implemented the method in Eq. 8 and derive the comparison results. Due to text length limitations, please refer to **Point 1 of our reply to Reviewer FFcG** for these details.
>
> Here are our responses and revisions:
>
> **2. Theoretical Clarifications:**
>
> 2.1 **Proof of Lemma 2:**
>    Thanks for your clarification on the rigor of the proof of Lemma 2. We have added the proof of the missing case when $|r_o| = l = 1$ here and also to the paper.
>
>
>    - If some edges are contained in $o$ and they form a consecutive path $(c_{r_1},c_{r_1+1},\cdots,c_{q_1})$. In this case, the rest part of the cycle $(c_{q_1},c_{q_1+1},\cdots,c_{r_1})$ is contained in the graph $G$. This forms a directed path from $c_{q_1}$ to $c_{r_1}$ for $(c_{r_1},c_{q_1})\in o^+$, which conflicts with the condition of the right-hand side of Equality (12) $X_j\leadsto X_i \notin G \text{ for }(i,j)\in o^+$. (Note that there is a typo in the original paper as $X_i\leadsto X_j \notin G \text{ for }(i,j)\in o^+$, which is fixed.)
>
> 2.2 **Sequential Ordering and Transitive Closure:**
>
> It seems that you concern about whether a graph satisfying $o$ equals satisfying its transitive closure $o^+$.
>
>    - $G$ satisfies partial order set $\mathcal{O}$ if and only if any path $X_j \leadsto X_i\notin G$ for all $(i,j)\in \mathcal{O}^+$ (Lemma 1, line 141). For a graph $G$ satisfying $\mathcal{O}$, we consider its transitive closure $\mathcal{O}^+$. Given that $(\mathcal{O}^+)^+$ is still $\mathcal{O}^+$, $G$ also satisfies $\mathcal{O}^+$.
>
>    - Besides, we derive Theorem 4 from Eq. 12 proved in Lemma 2 (line 538):
>
>      $G^\prime \in \text{DAG for } E(G^\prime) = E(G) \cup o \iff G \in \text{DAG and } X_j\leadsto X_i \notin G \text{ for }(i,j)\in o^+$
>
>      Note that the left-hand condition equals $h(\mathcal{A}(W,o))=0$. Combining with Eq. 11 (line 533), we derive:
>
>       $h^\prime (W,\mathcal{O})=0 \iff G\in \text{ DAG and } X_j \leadsto X_i \notin G \text{ for } (i,j)\in \cup_{o\in \mathcal{P}(\mathcal{O}^-)} o^+$
>
>      With the result $\cup_{o\in \mathcal{P}(\mathcal{O}^-)} o^+ = \mathcal{O}^+$ from Lemma 3, we have:
>
>      $h^\prime (W,\mathcal{O})=0 \iff G\in \text{DAG} \text{ and } X_j\leadsto X_i\notin G \text{ for }(i,j)\in \mathcal{O}^+$
>
>      The path absence condition on the right-hand side is equivalent to adherence to $\mathcal{O}$.
>
>    - As for Remark 3 (line 199), it is to help readers understand our identical representation for paths and partial orders, which does not influence the transitive nature of them. We will clarify this in the paper.
>
> Please let us know if you have further questions.
>
> **1. Clarify the Novelty of Theorems:**
>
>    Thanks for pointing out the novelty of Theorems. Theorem 1 is previously proven by Wei et al. [1]. We re-illustrate this Theorem from a pure graphical perspective to help understand the connection between orderings and path existences. Theorem 3 can be derived by the proof of Theorem 1. Your concern reminds us to prioritize the novelty; therefore, we will change the display of Theorems 1, 2, and 3 to Proposition 1, Proposition 2, and Corollary 3, and also explicitly clarify the reference to the previous work. Proposition 2 is a trivial result in graph theory, illustrated in the paper to help readers understand the typical complexity from integrating partial orders compared to total ordering, which will also be clarified in the paper. Lemma 1 functions an important result to bridge the ordering space and graph space, which is also a trival result in the Graph Theory and will be clarified in the paper.
>
>    [1] Wei, D., Gao, T., & Yu, Y. (2020). DAGs with No Fears: A closer look at continuous optimization for learning Bayesian networks. Advances in Neural Information Processing Systems, 33, 3895-3906.
>
> **3. Experimental Clarifications:**
>
> Yes, the run time is the time consumed for the algorithm to learn the provided solutions. The stopping condition is that the acyclicity loss $h$ reaches a threshold small enough to justify the acyclicity of the graph, which is set to $1 \times 10^{-8}$ in the experiment.
>
> **4. Trade-off Between Equations 8b and 9b:**
>
> Thanks for your interesting question. When all partial orders are graphically separate, meaning no sequential orderings longer than 1 exist in the partial order set, the two formulations have close numbers of terms (Eq. 9b has one fewer than Eq. 8b). In this case, the computational difference can be negligible. However, we still do not have an idea to construct a case where Eq. 8b is preferred. Please let us know if you have further considerations.
>
> **5. Example 1 Clarification:**
>
> Thanks for your question. You are correct that step 2 does not exist without step 1, which will be clarified in the paper.
>
> - The detailed process of our operation is to split the graphical structure of the transitive reduction $\mathcal{O}^-$ (as adherence to $\mathcal{O}^-$ equals adherence to $\mathcal{O}$) of the partial order set into paths (sequential orderings), and then add them individually to the acyclicity term to form the augmented acyclicity.
>
> - If we do not split the graphical structure of the partial order set into paths, the operation will directly add all the edges in $\mathcal{O}^-$ to the acyclicity term. This operation corresponds to the formulation $h(\mathcal{A}(W,\mathcal{O}^-))$, where the function $\mathcal{A}$ adds the edges in $\mathcal{O}^-$ to $W$.
>
>
> Thank you again for your insightful feedback, which has greatly helped in refining our paper. Please let us know if further clarifications are needed.

---

> > ### Comment · Reviewer_SXfo · 2024-08-11
> > **Acknowledgement of Author Rebuttal**
> >
> > I sincerely thank the authors for clarifying both my theoretical and empirical concerns. I now see that Theorem 4 indeed holds in its fullest generality and I would suggest the authors to replace lines 567-569 with the step-by-step explanation given in their answer as it is much clearer how Theorem 4 follows from Lemma 2 and 3. Additionally, the empirical comparison between Equation 8b and 9b given to Reviewer FFcG clearly shows the improvement in computational efficiency while maintaining performance. Finally, as also requested by Reviewer 99Ub, the authors have more clearly separated their novel result from previous results. Hence, I will increase my score to a full accept and congratulate the authors on the nice marriage of theory and practice.

---

> > > ### Author Response · Authors · 2024-08-11
> > >
> > > Thanks for your patient review and invaluable comments. We will refine related parts in alignment with your suggestion.

---

### Official Review · Reviewer_jpLj · 2024-07-12

**Soundness:** 3
**Presentation:** 2
**Contribution:** 4
**Rating:** 7
**Confidence:** 4

**Summary:**

The paper provides a solution for imposing partial ordering information into differentiable DAG learning, which is a very important problem. It also proposes an efficient implementation based on rigorous theoretical justification. With this prior information, even with fewer samples, better structural recovery can be achieved in experiments.

**Strengths:**

1. This paper imposes partial ordering into the differentiable DAG learning problem. It is a very interesting problem to the community.
2. The paper is theoretically well-supported.
3. This paper addresses computational issues when imposing partial ordering into differentiable DAG learning. The experiments indicate that better structural recovery can be achieved with the information of partial order.

**Weaknesses:**

1. I found there are some points in the paper that are hard to understand. A mild suggestion is to add examples to illustrate these concepts, which would make the paper more readable. For instance, toy examples to explain what $\mathcal{O}^+$ and $\mathcal{O}^-$ are. Especially in section 3.3, where there are many definitions, theorems, and lemmas, examples would help readers grasp the points and have a better understanding of the paper.
2. Typo: Line 204 $\mathcal{O}^-)$ should be corrected to $\mathcal{O}^-$.

**Questions:**

1. Lines 164-166: "Merely forbidding edges...not contained in $\mathcal{O}$" I'm not sure I understand this point. Could you provide some examples to illustrate this argument?
2. In Eq (6), $(i,j)\in \mathcal{O}^+$ indicates $i\prec j$, meaning there is no directed edge from $j$ to $i$. However, in Equation (8b), $(i,j)\in \mathcal{O}^+$ implies there is no directed edge from $i$ to $j$. Do I understand this correctly? If so, please make notation is consistent throughout the paper.
3. What Equation (9c) means here? $W - W\circ W_o$ can be regarded as remove edge in path $o$, then we add $\tau W_o$ back?
4. In the experiments, do you test $p$ is small, such as $p = 10$?

**Limitations:**

Yes

---

> ### Author Rebuttal · Authors · 2024-08-03
>
> Thank you for your detailed review and valuable questions. Here are our responses:
>
> **1. Explanation on the Difficulty of Integrating Partial Order Constraints**
>
> The statement "Merely forbidding edges that violate $ \mathcal{O}^+ $ is insufficient for compliance, as it is possible to *walk* from a variable to a preceding variable in $\mathcal{O}$ through another variable whose order with others is not contained in $\mathcal{O}$." highlights why simply constraining edge absence $(x_i,x_j)$ for all $(j,i)\in \mathcal{O}^+$ is **not enough** to fully satisfy partial order constraints.
>
> For example, consider four nodes $1,2,3,4$ with a partial order set $\mathcal{O}=\{(1,2),(2,3)\}$. We forbid all inverse edges in $\mathcal{O}^+$, which are $\{(2,1),(3,2),(3,1)\}$. Despite this, directed paths violating the partial order $(1,2)$ can still exist, such as the path $(2,4,1)$. Such paths can be constructed by traversing nodes not in $\mathcal{O}$, like node $4$ in this case. This demonstrates why integrating a total order is straightforward, while integrating partial orders is challenging due to variables whose order relations with others are unknown.
>
> **2. Typo Revision**
>
> Thank you for your comment. The typo in Eq. 8b has been noted. For a partial order $(x_i,x_j) \in \mathcal{O}^+$, there should not be any directed path from $x_j$ to $x_i$. We will thoroughly review all mathematical presentations to ensure correctness.
>
> **3. Explanation on Eq. 9c**
>
> You are correct about the operation in Eq. 9c. This operation aims to assign uniform weight $\tau$ to all virtual edges added to the augmented acyclicity term, indicating their existence. This serves two functions: **1) Avoid accidental edge removal:** The weight of an edge in $W$ can be negative, risking erroneous edge removal in the augmented acyclicity constraint and losing prior information. **2) Maintain stability:** The influence of the acyclicity term on edge recovery can be sensitive to weights, so we use uniform virtual weights to ease hyper-parameter selection. More details and analysis are provided in our reply to **Weakness 5 to Reviewer 99Ub**.
>
> **4. Supplementary Experiment with Small $p$**
>
> We conducted a supplementary experiment with $p=10$, indicating a small number of variables in the partial order sequence. Considering the random generation of partial orders from the topological ordering of a DAG, the partial order sequence may not address critical order relationships that contribute to better structure recovery when the variable number is small. For instance, if two nodes $x,y$ do not have an ancestor-descendant relation, then either $x \prec y$ or $y \prec x$ is correct, making the specification of such partial orders less impactful.
>
> We performed ten repetitions of the random generation of single-chained partial orders with $p=10$ (containing $0.1d$ variables in the sequence). The output quality results are reported individually for each repetition. Settings here are an ER-4 graph with a Gaussian noise model and the sample size $n=40$. Prior Na denotes the baseline without prior (NOTEARS). Cases where integrating the prior partial order yields better results than the baseline are highlighted in bold. Please note that all results presented are from a single repetition rather than an average of multiple repetitions.
>
> | Prior | Repeat | d=20 (SHD / F1)   | d=30 (SHD / F1)   | d=40 (SHD / F1)    | d=50 (SHD / F1)    |
> | ----- | ------ | ----------------- | ----------------- | ------------------ | ------------------ |
> | Na    | 1      | 68 / 0.43         | 98 / 0.48         | 122 / 0.52         | 165 / 0.46         |
> | $p=10$  | 1      | **64** / **0.46** | **93** / **0.53** | 136 / 0.47         | **160** / **0.49** |
> |       | 2      | 68 / **0.45**     | **93** / **0.52** | 125 / 0.51         | 175 / 0.45         |
> |       | 3      | **64** / **0.46** | **85** / **0.56** | **113** / **0.56** | 169 / 0.44         |
> |       | 4      | **66** / **0.46** | **94** / **0.52** | 137 / 0.45         | 170 / 0.46         |
> |       | 5      | 70 / 0.41         | **85** / **0.58** | 122 / **0.53**     | **162** / **0.49** |
> |       | 6      | **66** / **0.47** | 100 / **0.51**    | **121** / **0.53** | **152** / **0.53** |
> |       | 7      | **67** / 0.40     | **87** / **0.56** | 131 / 0.48         | 183 / 0.44         |
> |       | 8      | **63** / **0.48** | **78** / **0.60** | 126 / 0.51         | **160** / **0.51** |
> |       | 9      | 73 / 0.36         | **95** / **0.52** | 132 / 0.48         | **158** / **0.49** |
> |       | 10     | 70 / 0.40         | **94** / **0.50** | **119** / **0.54** | 165 / 0.46         |
>
> **Addressing Weaknesses**
>
> Thank you for the suggestion to improve the paper's clarity. We will add more examples to help readers understand the core concepts of the paper. Additionally, we will thoroughly check and correct all typos.

---

> > ### Comment · Reviewer_jpLj · 2024-08-11
> > **Thank you!**
> >
> > Thank you for addressing all of my concerns. At this point, I don’t have any additional concerns. I raise my score.

---

### Official Review · Reviewer_99Ub · 2024-07-18

**Soundness:** 3
**Presentation:** 3
**Contribution:** 2
**Rating:** 6
**Confidence:** 3

**Summary:**

This paper introduces an approach to integrate partial order constraints into differentiable structure learning for causal discovery. The key contributions are:
* Formulating an equivalent constraint set of path prohibitions to implement partial order constraints in the graph space.
* Proposing an efficient method to integrate partial orders by augmenting the acyclicity constraint.
* Proving the theoretical correctness and completeness of the proposed augmented acyclicity constraint.
* Demonstrating the effectiveness of the method through experiments on both synthetic and real-world datasets.

The authors show that their method can significantly improve the quality of recovered causal structures while maintaining computational efficiency, especially for long sequential orderings. They also demonstrate that using partial order constraints can reduce the required sample size for accurate causal discovery on real-world data.

**Strengths:**

* The paper addresses an important gap in differentiable structure learning by enabling the integration of partial order constraints.
* The augmented acyclicity approach handles long sequential orderings more efficiently, addressing a key limitation of a naive implementation.
* The experiments cover both synthetic and real-world datasets, demonstrating the method's effectiveness across various scenarios.
* Results on the Sachs dataset show significant improvements in structure recovery with reduced sample sizes, highlighting the method's impact for real-world applications.
* The proposed method is designed as a plug-and-play module that can be integrated with various differentiable structure learning algorithms.

**Weaknesses:**

* Several statements are known or follow easily from existing results.
* As noted in the limitations section, the method's efficiency can degrade with complex partial order structures involving multiple chains.
* The paper doesn't extensively discuss how sensitive the method is to incorrect or conflicting partial order priors, which is possible in applications.
* While the paper compares to baselines without priors, it doesn't compare to other methods that might incorporate different types of prior knowledge.
* The paper doesn't provide an in-depth analysis of how sensitive the results are to the choice of hyperparameters, particularly τ in the augmented acyclicity term.

**Questions:**

My overall take of this paper is that it could be a nice contribution as it appears to effectively handle partial ordering constraints for differentiable structure learning methods. However, some rewriting might be deemed necessary.

* The definition of the SEM is confusing, in eq.(1) you might want to define $z$ as well, and I also find eq.(2) confusing, the definition of W_kj is not a real number but then you are taking an inner product in the function for D_ij. First, why is it necessary to write the sample version? Second, it is clearer if you state what kind of models work for the results in the sequel.

* Several results are known or are trivial. Theorem 1 is already proved in Wei et al. (2020); Lemma 1 and Theorem 2 are straightforward results; Theorem 3 follows also directly from prior results in Wei et al. (2020). My point here being that theorems in a paper are typically representative of novel results and that correspond to the main contributions. In this case, I failed to see this. Theorem 4 should perhaps be Theorem 1 and the only theorem in the paper.

Minor things:
* L21: reform -> reformed
* L25: Ng et al., 2020, do not propose an aciclicity characterization.
* You can also consider citing Deng et al. 2023 ("Global Optimality in Bivariate Gradient-based DAG Learning") as they formally show, albeit in the bivariate case, how the continuous framework can succeed in recovering the underlying DAG.

**Limitations:**

Limitations are addressed in Appendix F.

---

> ### Author Rebuttal · Authors · 2024-08-03
>
> Thank you for your thorough review and valuable feedback. We begin by addressing your critical point regarding the analysis of the hyper-parameter $\tau$ as highlighted in Weakness 5. Following this, we will address your other questions and concerns. Here are our responses:
>
> **Weakness 5: Analysis of the Influence of Hyper-Parameter $\tau$**
>
> We agree that an in-depth analysis of the hyper-parameter $\tau$ is essential. $\tau > 0$ serves as the uniform weight for the edges in $\mathcal{P}(\mathcal{O}^-)$, which are added solely to the augmented acyclicity term $ h'(W,\mathcal{O}) $ and not to the data approximation term $ \mathcal{F}(W) $.  Here's our analysis of the sensitivity of results to $\tau$:
>
> - Keep in mind that $\tau$ represents the weight of *virtual* edges, which may not be present in the actual graph but are included in the graph used for enforcing the acyclicity constraint, referred to as the **acy-graph**.
>
> - **Small $\tau$**: When $\tau$ is too small, it fails to indicate the presence of virtual edges in the acy-graph effectively. This can make the absence of reversed paths not adequately enforced, losing prior information and even breaking acyclicity. In essence, a small $\tau$ can be seen as removing all edges in $\mathcal{P}(\mathcal{O}^-)$ from the acy-graph instead of adding these edges, thus weakening the acyclicity constraint and **leading to cycles**.
>
> - **Large $\tau$**: Conversely, a large $\tau$ can disrupt numerical stability by ignoring small weights that signify edge absences. For instance, we consider an edge to be absent if with a weight below a threshold $r$, and we have a forbidden length-3 cycle $ p = W_{1,2}W_{2,3}W_{3,1} $, setting $W_{1,2} = \tau$ to a large value while $W_{2,3} = r^-$ (indicating absence), can result in the influence of $W_{1,2}$ on $W_{3,1}$ being incorrectly propagated through backpropagation. Specifically, $\nabla_{W_{3,1}} p = \tau \times r^-$ becomes large enough to enforce the absence of $W_{3,1}$, even though this cycle is already absent due to the absence of $W_{2,3}$. This misinterpretation can lead to the erroneous removal of edges, potentially **resulting in an empty graph**.
>
> - We conducted experiments with varying $\tau$ values and observed the real acyclicity loss $ h $, augmented acyclicity loss $ h' $, data approximation loss $ \mathcal{F} $, and output's DAG condition, edge count, and F1 score. Results for an ER-4 graph with Gaussian noise model, node count $ d = 20 $, sample size $ n = 40 $, and edge threshold $ \gamma = 0.1 $ are reported here and align with our discussion.
>
>   | $\tau$        | 0.0  | 0.1  | 0.2  | 0.3      | 0.4      | 0.5      | 1.0      | 2.0      | 3.0      | 4.0      | 5.0  | 6.0  | 7.0   | 8.0  |
>   | ------------- | ---- | ---- | ---- | -------- | -------- | -------- | -------- | -------- | -------- | -------- | ---- | ---- | ----- | ---- |
>   | $h$           | 0.4  | 1e-4 | 0.2  | 2e-6     | 4e-7     | 1e-9     | 4e-9     | 1e-10    | 5e-11    | 6e-9     | 3e-9 | 7e-5 | 1e-12 | 0.0  |
>   | $h^\prime$    | 6e-9 | 2e-5 | 6e-4 | 2e-8     | 2e-8     | 3e-8     | 6e-9     | 5e-9     | 4e-9     | 2e-10    | 8e-8 | 3.9  | 1e-2  | 5e-2 |
>   | $\mathcal{F}$ | 9.6  | 5e+3 | 4e+3 | **12.9** | **12.7** | **16.9** | **11.7** | **11.7** | **11.5** | **11.7** | 8e+2 | 2e+4 | 3e+4  | 3e+4 |
>   | $\text{DAG?}$ | ❌    | ❌    | ❌    | ✓        | ✓        | ✓        | ✓        | ✓        | ✓        | ✓        | ✓    | ✓    | ✓     | ✓    |
>   | @Edge         | 53   | 31   | 55   | 61       | 62       | 69       | 71       | 70       | 66       | 67       | 33   | 1    | 0     | 0    |
>   | F1            | 0.27 | 0.27 | 0.36 | 0.61     | 0.65     | 0.56     | 0.68     | 0.69     | 0.70     | 0.69     | 0.23 | 0.0  | 0.0   | 0.0  |
>
> **Questions**
>
> - Thank you for pointing out the missing note regarding noise $z$ in the SEM and the confusion caused by the sample version definition. We have removed the sample version SEM definition and clarified the applicable models within the context, covering both linear and nonlinear models.
>
> - Thank you for pointing out the novelty issues with our statement presentation. We have clearly cited sources for statements not originally proposed. Theorems 1, 2, and 3 have been reclassified as propositions, with Theorem 4 now presented as the sole Theorem 1.
>
> - The typo on L21 has been corrected, and the citation for Ng et al., 2020 has been moved to the correct position.
>
> - Thank you for your suggestion. We find the work by Deng et al. (2023) highly relevant for demonstrating the theoretical promise of the continuous framework in DAG learning and have added it to the introduction section.
>
> **Addressing Other Weaknesses**
>
> - **Weakness 1**: Addressed above.
> - **Weakness 2**: Acknowledged in the limitations section.
> - **Weakness 3**: As noted in the Broader Impact section (line 656), the method is sensitive to prior errors due to its hard constraint on partial orders. However, based on our empirical experience with edge constraints, using a soft constraint alternative also has limitations in addressing prior errors and may reduce the benefits derived from prior information.
> - **Weakness 4**: Analyzing different types of priors is theoretically complex because they influence structure learning in varied ways. Most existing methods treat priors as secondary and struggle to handle complex priors beyond edge constraints. However, edges are typically the aim of discovery from data rather than being available as rich, pre-existing priors. Therefore, we believe a detailed analysis of different priors is beyond the scope of this paper.
>
> Thank you again for your insightful feedback. Please let us know if you have further concerns.

---

> > ### Comment · Reviewer_99Ub · 2024-08-13
> >
> > I thank the authors for their response. I will increase my score to 6 for now.

---

### Decision · Program_Chairs · 2024-09-25

**Decision:**

Accept (poster)

**Comment:**

This paper describes a method of incorporating partial order constraints into differential DAG learning. The authors devise a mechanism to incorporate these constraints into the acyclic constraints, highlight why partial ordering constraints pose significant difficulty, and develop a technique to impose the constraints in a more efficient manner.  After author discussion, reviewers were uniformly positive, viewing the method as novel and with practical significance.  Reviewers also offered several constructive suggestions to improve the presentation.